# Ice-nucleating particle versus ice crystal number concentration in altocumulus and cirrus embedded in Saharan dust: A closure study

Albert Ansmann[1], Rodanthi-Elisavet Mamouri[2], Johannes Bühl[1], Patric Seifert[1], Ronny Engelmann[1], Julian Hofer[1], Argyro Nisantzi[2], James D. Atkinson[3,a], Zamin A. Kanji[3], Berko Sierau[3,b], Mihalis Vrekoussis[4,5,6], and Jean Sciare[4]

[1]Leibniz Institute for Tropospheric Research, Leipzig, Germany
[2]Department of Civil Engineering and Geomatics, Eratosthenes Research Center, Cyprus University of Technology, Limassol, Cyprus
[3]Institute for Atmospheric and Climate Science, ETH Zürich, Zurich, Switzerland
[4]Energy, Environment and Water Research Center, The Cyprus Institute, Nicosia, Cyprus
[5]Institute of Environmental Physics and Remote Sensing, University of Bremen, Bremen, Germany
[6]Center of Marine Environmental Sciences (MARUM), University of Bremen, Bremen, Germany
[a]now at: Ethical Power, Exeter, UK
[b]now at: Notime, Zurich, Switzerland

*Correspondence to:* A. Ansmann
(albert@tropos.de)

**Abstract.** For the first time, a closure study of the relationship between ice-nucleating particle concentration (INPC) and ice crystal number concentration (ICNC) in altocumulus and cirrus layers, solely based on ground-based active remote sensing, is presented. Such aerosol-cloud closure experiments are required (a) to better understand aerosol-cloud interaction in the case of mixed-phase clouds, (b) to explore to what extent heterogeneous ice nucleation can contribute to cirrus formation which is usually controlled by homogeneous freezing, and (c) to check the usefulness of available INPC parameterization schemes, applied to lidar profiles of aerosol optical and microphysical properties up to tropopause level. The INPC-vs-ICNC closure studies were conducted in Cyprus (Limassol and Nicosia) during a six-week field campaign in March-April 2015 and during the 17-month CyCARE (Cyprus Clouds Aerosol and Rain Experiment) campaign. Focus was on altocumulus and cirrus layers which developed in pronounced Saharan dust layers at heights from 5-11 km. As a highlight, a long lasting cirrus event was studied which was linked to the development of a very strong dust-infused baroclinic storm (DIBS) over Algeria. The DIBS was associated with strong convective cloud development and lifted large amounts of Saharan dust into the upper troposphere, where the dust influenced the evolution of an unusually large anvil cirrus shield and the subsequent transformation into an cirrus uncinus cloud system extending from the Eastern Mediterranean to Central Asia, and thus over more than 3500 km. Cloud top temperatures of the three discussed closure-study cases ranged from $-20°C$ to $-57°C$. INPC was estimated from polarization/Raman lidar observations in combination with published INPC parameterization schemes, whereas ICNC was retrieved from combined Doppler lidar, aerosol lidar, and cloud radar observations of the terminal velocity of falling ice crystals, radar reflectivity and lidar backscatter in combination with modeling of backscattering at 532 nm and 8.5 mm wavelength. Good to acceptable agreement between INPC (observed before and after the occurrence of the cloud layer under investigation) and ICNC values was found in the discussed three proof-of-concept closure experiments. In these

case studies, INPC and ICNC values matched within an order of magnitude (i.e., within the uncertainty ranges of the INPC and ICNC estimates), and ranged from 0.1–10 $L^{-1}$ in the altocumulus layers and 1-50 $L^{-1}$ in the cirrus layers observed between 8–11 km height. The successful closure experiments corroborate the important role of heterogeneous ice nucleation in atmospheric ice formation processes when mineral dust is present. The observed long-lasting cirrus event could be fully explained by the presence of dust, i.e., without the need for homogeneous ice nucleation processes.

## 1   Introduction

Heterogeneous ice formation is an important pathway of aerosol-cloud interaction. Via heterogeneous freezing and nucleation mechanisms solid aerosol particles trigger the nucleation of ice crystals at relatively high temperatures from 0°C to about $-35$°C. At these temperatures, ice formation (and the formation of mixed-phase clouds) is not possible without the assistance of ice-nucleating particles (INPs) (Pruppacher and Klett, 1997; Kärcher and Seifert, 2016). Even at lower temperatures ($-40$ to $-65$°C), at which homogeneous freezing usually dominates, heterogeneous ice nucleation on dust and soot particles can have a strong impact at relative humidities over ice of $RH_i < 140$-150% on the evolution and life time of tropospheric clouds. Because the occurrence of the ice phase is an important prerequisite to initiate the formation of precipitation (almost everywhere around the globe) (Mülmenstädt et al., 2015), aerosol particles, when acting as INP, can have a sensitive impact on the vertical distribution of water vapor and cloud water and thus on the tropospheric water cycle.

Heterogeneous ice nucleation is however not well understood due to the complexity of ice-nucleating efficiency of atmospheric aerosol mixtures depending on their specific chemical and microphysical properties (Hoose and Möhler, 2012; Murray et al., 2012; Kanji et al., 2017) and the complex influences of ambient temperature, humidity, air motion (updrafts, downdrafts, turbulence and entrainment) and associated ice supersaturation conditions on the nucleation of ice crystals (Lohmann et al., 2016). In cirrus layers, the competition between heterogeneous and homogeneous ice nucleation further complicates the effort to improve our knowledge about the role of aerosols in ice formation processes in the climate system (Kärcher et al., 2014; Kärcher, 2017; Kuebbeler et al., 2014; Jensen et al., 2016). As a consequence, aerosol-cloud interaction remains an important source of large uncertainty in weather and climate predictions.

Despite the gaps in our knowledge about heterogeneous freezing, tremendous progress has been made in the field of mixed-phase and ice cloud research. Numerous aerosol types and components have been characterized in the laboratory and field campaigns regarding their ice nucleating potential (see the review of Kanji et al., 2017). Recent field activities with focus on ice-nucleating particle concentration (INPC) were reported by Schrod et al. (2017); Welti et al. (2018); McCluskey et al. (2017, 2018a, b); Price et al. (2018) and Wex et al. (2018). More than 30 years of airborne observations in mixed-phase and cirrus clouds improved our understanding of ice formation processes and have provided a wealth of information on the microphysical properties of ice clouds (see the review articles of Heymsfield et al., 2017; Field et al., 2017; Korolev et al., 2017). Prenni et al. (2009) and Eidhammer et al. (2010) were, to our knowledge, the first who successfully performed closure studies on the relationship between INPC and ice crystal number concentration (ICNC), based on airborne in situ observations. Such closure studies (see definition in Sect. 2) are one of the most direct and powerful efforts to investigate the aerosol impact on the

development of the ice phase in clouds. Costa et al. (2017) also made an attempt to combine aerosol and ice crystal information gained from airborne in situ observations to characterize the link between INPC and ICNC and thus the potential impact of heterogeneous ice formation on cirrus evolution.

However, it remains difficult to obtain a clear picture of the influence of aerosols on the life cycle of ice-containing clouds from airborne in situ measurements. Usually the environmental conditions (temperature, relative humidity, INPC) at which the ice crystals nucleated remain unkown in the analysis of aircraft observation performed within the clouds. Most of the aircraft tracks are hundreds of meters below cloud top and thus below the coldest region of the cloud where the probability of ice nucleation is highest. After nucleation, the ice crystals grow fast to sizes of 50–100 $\mu$m within minutes (Bailey and Hallett, 2012) and immediately start falling through the cloud deck and influence the further evolution of the entire cloud system from the top to base and the virga zone (Spichtinger and Gierens, 2009a, b; Field and Heymsfield, 2003). Clear and unambiguous conclusions on the specific impact of aerosol particles on the evolution of the ice phase can only be obtained by monitoring aerosol layering and embedded cloud systems at all heights simultaneously from cloud base to top.

A promising way to explore cloud evolution processes with focus on heterogeneous ice nucleation is the use of ground-based remote sensing (see, e.g., the approach presented by Simmel et al., 2015). Continuous vertical profiling of aerosol, altocumulus and cirrus layers (from the base of the virga zone up to cloud top at the same time) with lidars and radars allows us to study processes rather coherently and in large detail. As illustrative examples we presented a Raman-lidar-based documentation of the impact of a strong gravity wave on ice production in a mixed-phase altocumulus layer (Ansmann et al., 2005) and reported the complete life cycle of a tropical altocumulus layer from the birth to the decay of the cloud system by complete glaciation at $-34°C$ monitored with wind Doppler lidar and cloud-phase-resolving polarization lidar (Ansmann et al., 2009).

Recently, retrieval techniques have been introduced to estimate ICNC (Sourdeval et al., 2018; Mitchell et al., 2018; Bühl et al., 2019) and INPC (Mamouri and Ansmann, 2015, 2016; Marinou et al., 2019) from spaceborne and ground-based lidar and radar observations. The potential is therefore given to cloud-process-resolving as well as global scale information and large statistics on the link between INPC and ICNC and thus to quantify the impact of aerosol particles on the evolution of, e.g., cirrus clouds on a global scale (Gryspeerdt et al., 2018). However, active remote sensing with polar-orbiting satellites deliver snapshot-like observations of the atmospheric state only and thus do not permit the investigation of the fundamental aerosol-cloud interaction processes in detail such as the birth and evolution (life cycle) of a cloud system (over minutes to hours) in a given aerosol environment. Passive remote sensing by means of geostationary satellite, on the other hand, can provide cloud information with high temporal resolution but not with sufficient height resolution.

The idea to perform INPC-vs-ICNC closure studies, solely based on active ground-based remote sensing, came up after the introduction of a new lidar method to estimate INPC profiles throughout the troposphere (Mamouri and Ansmann, 2015, 2016). However, the additionally required ICNC retrieval technique was only recently developed (Bühl et al., 2019). Both methods are briefly outlined in Sect. 4. Before, we introduce our field campaigns in Sect.3. We applied the remote-sensing-based INPC-vs-ICNC closure approach for the first time during the Cyprus-2015 campaign conducted in the framework of the BACCHUS (Impact of *B*iogenic versus *A*nthropogenic emissions on *C*louds and *C*limate: towards a *H*olistic *U*nder*S*tanding, https://www.bacchus-env.eu/) project. Later, we continued with closure studies during the 17-month CyCARE (*Cy*prus *C*loud

*Aerosol* and *Rain Experiment*) campaign performed at Limassol from October 2016 to March 2018. Sect. 5 contains the results of the selected three closure studies which may be regarded as proof-of-concept studies. Two cases with prevailing deposition nucleation in thin and thick cirrus layers and one case with immersion freezing in an altocumulus layer are discussed. Summarizing and concluding remarks are given in Sect. 6. As a highlight, we will discuss our observation of the final phase of

a unique, unusually long-lasting cloud life cycle (Sect. 5.2, case study 2, 17 March 2015). The event started with the generation of a dust-infused baroclinic storm (DIBS) (Fromm et al., 2016; Caffrey et al., 2018) that was associated with strong cloud convection over desert areas of northern Africa followed by the evolution of rather large anvil cirrus shield (of more than 15000 km$^2$) and cirrus uncinus fields that extending from the central Mediterranean to central Asia (over more than 3500 km). The storm lifted large amounts of Saharan dust into the upper troposphere and thus created favorable conditions for a strong

contribution of heterogeneous ice nucleation to ice production and apparently to an extension of cirrus life time. This finding seems to be in contradiction with the established hypothesis that heterogeneous ice formation usually leads to a reduction of cirrus life time (e.g., Storelvmo et al., 2018; Gruber et al., 2019).

## 2   Short review on cloud closure experiments in the case of mixed-phase and ice clouds

INPC-vs-ICNC closure studies are of key importance in our effort to better understand and quantify the specific impact of

aerosol particles (i.e., of mixtures of different aerosol types of different number concentrations) on the evolution of the cloud ice phase. Therefore, we begin with an historical overview of published (pioneering) attempts to simultaneously measure INPC and ICNC and to perform ice-nucleation-related aerosol-cloud closure studies in this section.

The concept of aerosol-related closure experiments (Russel et al., 1979; Bates et al., 1998; Russell and Heintzenberg, 2000; Ansmann et al., 2002) was originally introduced to investigate the complex relationships between physical, chemical, optical,

and radiative properties of atmospheric aerosol particles and thus to study the direct aerosol effect on climate based on aerosol observations (Quinn et al., 1996). In closure experiments, the measured value of a dependent variable is compared with the modeled or predicted value that is calculated from measured values of independent variables by using an appropriate model. The outcome of a closure experiment provides a direct evaluation of the combined uncertainty of the model and the measurements. If we transfer this concept to the field of heterogeneous ice nucleation, we may use ICNC as the dependent, measured

variable. Then, INPC may be regarded as the predicted ICNC value assuming that all INPs become ice crystals during favorable heterogeneous ice nucleation conditions and further assuming that secondary ice production processes (Hallett and Mossop, 1974; Field et al., 2017; Sullivan et al., 2017, 2018; Korolev et al., 2019) and homogeneous freezing events do not occur (i.e., are suppressed because of unfavorable meteorological conditions). INPC is calculated from measured independent variables such as particle surface area concentration, denoted as $s$, or particle number concentration of large particles with

radius >250 nm, denoted as $n_{250}$. The variables $s$ and $n_{250}$ are input parameters in INPC parameterization schemes (DeMott et al., 2010; DeMott et al., 2015; Ullrich et al., 2017) which are the models in our closure experiments. Details of the INPC computation from lidar observations are given in Mamouri and Ansmann (2016) and Marinou et al. (2019), and in Sect. 4.

The first attempts to find agreement between INPC and ICNC levels in natural cloudy environments date back to the early 1960s. Auer et al. (1969) performed closure studies by comparing INPC with ICNC observation in cap clouds (at 3350 m height above sea level, a.s.l., Elk Mountain, Wyoming, in autumn 1967). Hobbs (1969) reviewed even earlier INPC-vs-ICNC studies and investigated the link between INPC and ICNC in natural clouds at 2025 m height in the Olympic Mountains, Washington, in March 1968. Both field observations revealed a steady decrease of the ICNC/INPC ratio from values of >5000 at temperatures of −5 to −10°C towards values of the order of 1–10 at temperatures around −25°C. The high ICNC/INPC ratios >5000 were the result of secondary ice formation.

First ice closure studies based on airborne measurements were conducted about 10-15 years ago. As pointed out by Avramov et al. (2011), Prenni et al. (2009) re-analyzed airborne observations of INPC collected during an Arctic field campaign in northern Alaska during October 2004 and compared them against corresponding ICNC observations in long-lived stratiform mixed-phase Arctic clouds at temperatures < −10°C. Taking into account the uncertainty ranges for both types of measurements, the reanalysis showed that INPC and ICNC were approximately equal if only large ice particles (with sizes > 125 $\mu$m) were considered. For smaller ice particles, INPC and ICNC differed by more than two orders of magnitude. This difference was attributed to ice shattering artifacts (Korolev et al., 2011). However, ice multiplication effects caused by riming and ice-breakup processes cannot be fully excluded at these temperatures (Sullivan et al., 2017, 2018).

Eidhammer et al. (2010) then conducted an aircraft campaign on the close link between measured ICNC and INPC concentrations in stable, well-defined orographic clouds at 7000–7700 m height a.s.l. (temperatures in the clouds were between −20 and −30°C) over Colorado and Wyoming in November and December 2007. The measurements showed that observed and parameterized INPC values compared well in number with ice crystals observed to nucleate in the same cloud. ICNC ranged from 0.1 to 2 L$^{-1}$, and INPC were mostly between 0.05 and 1 L$^{-1}$. INPC/ICNC ratios from 0.7 to 9 were found.

Avramov et al. (2011) presented a complex INPC-vs-ICNC closure attempt by combining airborne in situ measurements (of INPC and ice crystal microphysical properties), active remote sensing with several cloud radars (cloud profiling in terms of radar reflectivity, Doppler velocities related to air motion and falling ice crystals), radiosonde observations of atmospheric state parameters, and large-eddy-simulation (LES) modeling to test the hypothesis that INPC values measured above cloud top can account for the observed ICNC values. Reasonable agreement between INPC and ICNC was obtained in this way for low-lying mixed-phase Artic cloud layers observed in northern Alaska in April 2008. However, a key aspect was that a second INP reservoir, located below the well-mixed cloud layer, had to be assumed in the LES modeling efforts as well. These INPs were slowly mixed upward (into the cloud) in the simulations and helped to maintain the INPC values in the model close to those observed. A similar approach, based on cloud radar observations and estimates of INPC from airborne in situ measurements of aerosol size distributions below and above a long-lasting shallow mixed-phase cloud deck over United Kingdom, was used in the investigation of a long-lived altocumulus layer regarding the impact of INPC on the cloud lifetime (Westbrook and Illingworth, 2013).

In strong contrast to these Arctic and mid-latitude aerosol-cloud closure studies at clean-air and thus low aerosol concentration levels, our remote-sensing-based INPC-vs-ICNC closure approach deals with stratiform altocumulus and cirrus layers which developed in pronounced Saharan dust layers in the middle and upper troposphere over Cyprus (at Middle-East me-

teorological conditions). Such clouds were frequently observed during our field campaigns, conducted in each of the spring seasons from 2015-2018, and thus are obviously very common in the Eastern Mediterranean. Cloud top temperatures typically ranged from $-20°C$ to about $-60°C$. Secondary ice production (SIP), that can sensitively disturb any ice closure experiment, is assumed to have a minor impact on ICNC at these low temperatures. The Hallett-Mossop SIP process is associated with

splinter ejection during riming of ice cyrstals at temperatures between $-3°C$ and $-8°C$ (Hallett and Mossop, 1974).

## 3 Cyprus field campaigns

The Mediterranean Basin is well recognized by IPCC (International Panel for Climate Change) (Stocker et al., 2013) as a hot spot for climate change, the impacts of which are expected to amplify further in the years to come. However, IPCC also identified aerosol-cloud-precipitation relationships as one of the unsolved problems of atmospheric research and thus their

simplified representation in atmospheric circulation models as one of the most prominent reasons for the large uncertainties in the present future-climate-change debate. The European Commission responded to this issue by funding the BACCHUS project. BACCHUS aimed at a better understanding of heterogeneous ice formation in tropospheric clouds around the globe and improved consideration of cloud processes in cloud-resolving and Earth system models. In the framework of this research initiative, in which about 20 European research institutes, universities, and weather services were involved, a series of short-

term intensive field observations was conducted in Cyprus in each spring of 2015–2017. The spring season coincides with the end of the rain season (November to March) in the Eastern Mediterranean and, at the same time, frequent dust outbreaks towards Cyprus from the deserts in northern Africa and the Middle East take place. In addition, the 17-month monitoring campaign CyCARE (covering two rain seasons) was performed in Limassol, Cyprus, from October 2016 to March 2018.

Cyprus in the center of the Eastern Mediterranean region offers favorable conditions for atmospheric and climate research,

especially in the field of cloud and precipitation formation with focus on the influence of natural (desert dust, soil dust, marine particles) and anthropogenic aerosols (urban haze, biomass burning smoke) on these processes. The island exhibits Middle East atmospheric and climate conditions and air quality is strongly affected by a mixture of urban haze, originating mainly from urban and industrial conglomerations in southeastern Europe, but also from Middle East and northern Africa, of biomass burning smoke from the North (e.g., Black Sea countries), mineral dust originating from arid regions in Turkey and Middle

East deserts, and Saharan dust (Lelieveld et al., 2002; Nisantzi et al., 2014, 2015; Abdelkadar et al., 2015; Mamouri et al., 2016; Pikridas et al., 2018; Michaelides et al., 2018). There are very few locations on Earth which experience such complex aerosol structures, vertical layering and mixtures which can sensitively influence cloud evolution and precipitation processes.

### 3.1 Cyprus 2015

As a first BACCHUS field experiment, the joint BACCHUS/ENVI-Med/ChArMEx Cyprus-2015 campaign took place from

4 March to 7 April 2015 and was coordinated by the The Cyprus Institute (CyI). The campaign was organized in collaboration with the French ChArMEx (Mallet et al., 2016) and ENVI-Med (see http://www.mistrals-home.org, ENVI-Med: Environmental Mediterranean program, project CyAr: Cyprus Aerosols and Gas Precursors) research initiatives. The Chemistry-Aerosol

Mediterranean Experiment (ChArMEx; http://charmex.lsce.ipsl.fr) (Mallet et al., 2016) is a collaborative research program federating international activities to investigate Mediterranean regional chemistry-climate interactions. The ENVI-Med regional program is a French cooperation initiative for countries in the Mediterranean Basin designed to encourage and strengthen high-level scientific and technological cooperation in the region as well as research networking on sustainable development and understanding the environmental (ENVI) operation of the Mediterranean Basin (Med). In partnership with France, the program is focused on Mediterranean rim countries.

In-situ observations of gaseous and particulate pollution including INPC, CCNC, particle size distribution, number concentrations, chemical composition, and particle optical properties (absorption, light scattering, extinction coefficients) were performed at the remote site of CAO (Cyprus Atmospheric Observatory, http://www.cyi.ac.cy/index.php/cao.html) at Agia Marina Xyliatou (35.0° N, 33.1° E, about 500 m a.s.l., see Fig. 1) in the northern part of the Troodos mountains about 25 km west of Nicosia, the capital city of Cyprus. UAVs (Unmanned Aerial Vehicles) were operated at the Agia Marina site to measure aerosol microphysical, optical and cloud-relevant properties as a function of height up to 2 km a.s.l. (Calmer et al., 2019).

The remote sensing facility was deployed on the roof of a CyI building which is located in the southern part of the Capital Nicosia (35.2° N, 33.4° E, 180 m a.s.l.). The different field sites are shown in Fig. 1. The instrumentation was run continuously over a six-week period. The station consisted of a multiwavelength polarization/Raman lidar Polly (*P*ortab*L*e *L*idar s*Y*stem, http://polly.tropos.de) (Engelmann et al., 2016; Baars et al., 2016) from the National Observatory Athens (NOA) (Marinou et al., 2019), a wind Doppler lidar (HALO Photonics) from the Leibniz Institute for Tropospheric Research (TROPOS) (Bühl et al., 2016), Leipzig, and a sun/lunar photometer (Cimel, CE318-T) of TROPOS, belonging to the Aerosol Robotic Network (AERONET) (Holben et al., 1998; Barreto et al., 2017). Another polarization/Raman lidar and AERONET sun photometer was operated by the Cyprus University of Technology (CUT-TEPAK in Fig. 1) at Limassol (Mamouri et al., 2013).

## 3.2 Cyprus 2016

The follow-up Cyprus-2016 campaign included further aerosol and INP observations performed by the German INUIT (Ice Nuclei Research Unit, https://www.ice-nuclei.de/the-inuit-project/) research group. The INUIT consortium studied heterogeneous ice formation in the atmosphere within three different work packages divided into laboratory studies, field measurements, and modeling. The European research infrastructure project ACTRIS-2 (Aerosols, Clouds, and Trace gases Research InfraStructure, https://www.actris.eu/) supported the field activities in Cyprus as well. Almost the same infrastructure consisting of a ground station (Agia Marina Xyliatou), an UAV airport (now at Orunda, 7 km northeast of Agia Marina Xyliatou, and 21 km west of the lidar station at Nicosia), and the NOA Polly site (Marinou et al., 2019) at the CyI premises was available during the Cyprus-2016 campaign. The Doppler lidar of TROPOS was however not deployed in 2016.

For the first time, an in-depth comparison of aerosol mass profiles, INPC and INP-relevant aerosol parameters obtained from UAV flights and derived from the lidar observations were performed during the Cyprus 2016 campaign (Schrod et al., 2017; Mamali et al., 2018; Marinou et al., 2019). Some of these results will be discussed in Sect. 5.

## 3.3 CyCARE (2016–2018)

As one of the central BACCHUS remote sensing initiatives the mobile Leipzig Cloudnet supersite LACROS (Leipzig Aerosol and Cloud Remote Observation System, http://lacros.rsd.tropos.de/) (Bühl et al., 2013, 2016) was moved to the Eastern Mediterranean and was run at CUT in the city center of Limassol (34.7°N, 33°E, see Fig. 1), Cyprus, from 22 October 2016 to 26 March 2018, in the framework of CyCARE (Cyprus Aerosol, Cloud and Rain Experiment). The CyCARE campaign is part of a long-term cooperation between TROPOS and CUT established in 2012 and integrated into the Cloudnet activities coordinated by the European Union infrastructure project ACTRIS-2. The mobile Leipzig Cloudnet supersite is shown in Fig. 2 and is equipped with a multiwavelength polarization/Raman lidar Polly, wind Doppler lidar, 35 GHz Doppler cloud radar, ceilometer, disdrometer, and microwave radiometer. All tools were run continuously over the 17-month period. In addition, nearby AERONET sun photometer observations (CUT-TEPAK site) were taken. An excellent data set was collected for in-depth studies of the impact of dust and aerosol pollution on cloud and rain evolution in the Eastern Mediterranean. During the intensive CyCARE field phase in April 2017, 43 Vaisala radiosondes were launched at Limassol (Dai et al., 2018).

## 4  INPC and ICNC from ground-based active remote sensing

Figure 3 illustrates our approach of a cloud closure experiment with focus on heterogeneous ice nucleation. INPC is estimated from lidar observations in a cloud-free region before or after the passage of the cloud field. INPC values at cloud top height level are compared with ICNC values obtained from combined backscatter lidar, wind Doppler lidar, and cloud Doppler radar observations in the lower part of a cirrus cloud (see Fig. 3a) or in the virga zone below a mixed-phase altocumulus layer (see Fig. 3b). We assume that ice crystals nucleate at the coldest point of the cloud (at cloud top), where INPC is highest because of the rather strong INP number increase with deceasing temperature (Kanji et al., 2017). Furthermore, we assume that the freshly formed ice crystals grow fast by water vapor deposition, immediately start falling and do not collide and form aggregates so that the number of ice crystals does not change during sedimentation to the lower part of the cloud and the virga zone. In Sect. 4.1, we briefly outline the methods of the INPC retrieval (Mamouri and Ansmann, 2016; Marinou et al., 2019). Section 4.2 deals with the estimation of ICNC.

The uncertainties in our closure methodology caused by the idealized assumptions may be small in the case of geometrically and optically thin ice clouds (case study 1 in Sect. 5.1) with a vertical extent of a few 100 meters and in case of shallow mixed-phase altocumulus layers (case study 3 in Sect. 5.3) with the nucleation of a comparably low amount of ice cyrstals within the thin liquid-water layer at the top of the altocumulus. However, in geometrically and optically thick cirrus (as is the case in case study 2, Sect. 5.2) the assumption of an height-independent ice crystal number concentration from the top, at which ice crystals preferably nucleate, down to the virga zone is probably not well justified. Crystal-crystal collisions and subsequent aggregation processes can cause a significant decrease by a factor of 3-10 in ICNC from the upper part to the lower part of a cirrus deck (Field and Heymsfield, 2003; Field et al., 2006). Based on CALIPSO (Cloud-Aerosol Lidar and Infrared Pathfinder Satellite Observations), the retrieved ICNC near cloud top was frequently five times higher than in the lower half of vertically deep cirrus clouds (Mitchell et al., 2018). Further processes such as ice nucleation within the central and lower parts of the

cirrus layer as well as dilution and accumulation effects by turbulent processes and wind shear can lead to a strongly varying ICNC profile from cloud top to base (Spichtinger and Gierens, 2009a, b). Since we concentrate on clouds and heterogeneous ice nucleation at temperatures $< -20°$C secondary ice nucleation (Field et al., 2017; Korolev et al., 2019) may not introduce further uncertainties.

## 5   4.1   INPC retrieval

INPC profiling with lidar dates back to 2006 (Ansmann et al., 2008) when we made an attempt to investigate the impact of dust particles on ice formation in altocumulus which formed at the top of the Saharan dust layer over southeastern Morocco at temperatures from $-10$ to $-18°$C. Later, when aerosol-based INP parameterization schemes became available (DeMott et al., 2010; DeMott et al., 2015; Niemand et al., 2012; Ullrich et al., 2017), a lidar-based methodology was developed (Mamouri and Ansmann, 2015). In the first step of the INPC estimation, height profiles of the dust-related extinction coefficient $\sigma_d$ at 532 nm wavelength and the non-dust extinction coefficient $\sigma_c$ for continental aerosol particles (anthropogenic haze, biomass burning smoke, biological decay products) are derived and then converted into number concentrations of large aerosol particles $n_{250,d}$ and $n_{250,c}$ (considering particles with radius >250 nm only) and particle surface area concentrations $s_d$ and $s_c$. These microphysical parameters together with temperature profiles are used as input for the INPC estimation by means of INPC parameterization schemes as given in Table 1. We use GDAS (Global Data Assimilation System) temperature profiles of the National Weather Service's National Centers for Environmental Prediction (NCEP) in our computations. NOAA's Air Resources Laboratory (ARL, https://www.ready.noaa.gov/gdas1.php) NCEP model GDAS1 output archives contain these data (GDAS, 2019). During CyCARE we found remarkably good agreement between the GDAS1 and radiosonde temperature profiles (Dai et al., 2018). The standard deviation (root-mean-square deviation based on all 43 radiosonde profiles and respective GDAS1 profiles) was 0.87 K.

The complete lidar-based INPC retrieval is decribed in detail by Mamouri and Ansmann (2016). The uncertainties in the products are as follows: The dust extinction coefficients can be obtained with 15-25% relative error, the non-dust extinction coefficients with an uncertainty of 20-30%. The uncertainty in the microphysical parameters are of the order of 20-30% for the dust component and 25-40% for the non-dust aerosol fraction. The uncertainty in the INPC values is finally of the order of a factor of 2–5. The range of uncertainty around the estimated INPC values is thus one order of magnitude (Mamouri and Ansmann, 2016; DeMott et al., 2017; Marinou et al., 2019) when using published INP parameterizations.

Favorable INPs are insoluble particles such as mineral dust particles, biological material, volcanic ash and dust, and soot particles originating from open fires, burning and heating processes (e.g., Seifert et al., 2010, 2011, 2015; Hoose and Möhler, 2012; Murray et al., 2012; Kanji et al., 2017). According to the INP parameterizations in Table 1, we consider immersion freezing (D10, D15, U17-I) and deposition nucleation (U17-D). In the case of immersion freezing, ice nucleates on a solid particle immersed within a supercooled liquid droplet (e.g. Murray et al., 2012; Vali et al., 2015). Particles may contain insoluble and soluble components. The soluble fraction triggers the formation of supercooled liquid droplets, and the insoluble part is then responsible for heterogeneous freezing events. The D10 parameterization can be used to estimate INPC from the measured overall (dust plus non-dust) aerosol number concentration of particles with radius >250 nm. We use the D10

retrieval method also for dust and non-dust INP estimation in Sect. 5. The D15 parameterization was explicitly introduced for dust particles. Deposition nucleation describes the process when an ice embryo directly forms by water vapor deposition to an insoluble surface. The deposition nucleation parameterization (U17-D in Table 1) includes pore condensation and freezing (PCF) (Marcolli, 2014) occurring in voids and cavities of aggregated primary particles at relative humidity over water of RH$_w$ <100% and low temperatures of $< -38°$CK when at least one pore is filled with water.

Our INPC-vs-ICNC closure study however ignores contact nucleation and pre-activation influences. Contact freezing occurs when an INP initiates freezing by colliding with a supercooled droplet (see latest research by Hoffmann et al., 2013). As outlined in detail by Marcolli (2017), pre-activation denotes the capability of particles or materials to nucleate ice at lower relative humidities or higher temperatures compared to their intrinsic ice nucleation efficiency after having experienced an ice nucleation event before. The subsequent, next ice nucleation event is then thought to occur because of ice preserved in pores between the first ice nucleation event and second ice growth cycles.

There is an ongoing discussion on the applicability of the INP parameterization methods (Phillips et al., 2013; Boose et al., 2016; DeMott et al., 2017; Schrod et al., 2017; Price et al., 2018; Marinou et al., 2019). Our closure study contributes to this discussion. Because most of the INP parameterization schemes are based on laboratory studies performed at well-defined meteorological conditions (temperature, cooling rate, ice supersaturation level) and precisely known aerosol properties (aerosol type, chemical composition, size distribution), the central question arises: Can these parameterisations be applied to predict INPC, e.g., in the upper troposphere up to cirrus height level, at which aged, chemically- and cloud-processed aerosol particles may prevail? Chemical reactions on the surfaces of the particles may have already significantly changed the potential of the particles to serve as ice nuclei (Archuleta et al., 2005; Möhler et al., 2008; Cziczo et al., 2009; Sullivan et al., 2010a, b; Wex et al., 2014).

As a first activity of the Cyprus-2015 data analysis, we checked our INP retrieval approach by comparing the lidar-derived INPC values with in situ INPC observations taken at Agia Marina Xyliatou during the Cyprus-2015 spring campaign. Results are shown in Sect. 5. Marinou et al. (2019) furthered the discussion on the applicability of the D10, D15, U17-I,and U17-D paramaterizations by comparing lidar-derived INPC profilers with UAV-based in situ observations performed during the Cyprus-2016 campaign (Schrod et al., 2017). These results are also discussed in Sect. 5.

## 4.2   ICNC retrieval

Three different ways are used in order to derive ICNC from combined lidar and radar observations. The basic methodology is described in detail in Bühl et al. (2019) and briefly in this section.

### 4.2.1   Combined observations of cloud radar reflectivity and ice extinction coefficient from backscatter lidar

Lidar return signals (backscatter) and cloud radar reflectivity show approximately a diameter (D) and (equivalent) diameter dependence on ice crystal size of $D^2$ and $D^6$, respectively. This difference in sensitivity between both signals can be exploited in order to derive information about particle size. ICNC is estimated by comparing simulations of the ice crystal light-extinction coefficient at 532 nm and of the radar reflectivity at 8.5 mm wavelength with the respective measured extinc-

tion coefficients (Polly lidar) and radar reflectivity values. The lidar extinction values are obtained from the observed cirrus backscatter coefficients after multiplication with the climatological mean cirrus extinction-to-backscatter ratio of 32 sr (Seifert et al., 2007; Giannakaki et al., 2007; Josset et al., 2012; Garnier et al., 2015; Haarig et al., 2016). In the simulations, ICNC and the crystal size distribution are input parameters. For realistic size distributions ICNC is varied until the simulations match the observations. Such a combined lidar/radar approach has been widely used before, e.g., for estimating particle properties from spaceborne lidar and radar (DARDAR) (Ceccaldi et al., 2013). Combined lidar/radar observations of LACROS are available for the complete CyCARE campaign.

### 4.2.2 Observations of ice crystal terminal fall velocity spectrum (Doppler cloud radar), radar reflectivity, and extinction coefficient (backscatter lidar)

The ice crystal size distribution can be estimated via measurements of the terminal fall velocity spectrum with Doppler cloud radar. The mean terminal fall velocity $v_t$ of the ice crystals is difficult to derive because it is usually offset by the vertical motion of the air. Recently, methods have been developed to measure terminal fall velocity directly (Radenz et al., 2018), but those require additional instrumentation not available in the Cyprus experiments. However, the involved Doppler lidar detects both the Doppler velocity of falling ice crystals and of small liquid droplets moving with air parcel velocity in the cloud top layer. Both information together is used to estimate the impact of updraft and downdraft velocities on the retrieved ice-crystal terminal velocities in the ice virga below the mixed-phase cloud layers. In a similar way, the Doppler lidar is also able to detect updrafts and downdrafts in the cirrus top region where small ice crystals showing only small vertical velocities dominate. Again, this information is used to correct for air motions (eventually induced by gravity waves, radiative cooling and entrainment processes). In the case of stratiform mixed-phase clouds with shallow liquid-water layer at cloud top, $v_t$ is usually offset by the vertical air motion $v_{air}$, which is estimated at the base of the liquid-water layer directly above the zone dominated by ice crystals. In the ICNC retrieval, we take into account that values of $v_t$ from Doppler lidar and cloud radar are weighted by the particle area, or the particle mass squared, respectively. The terminal velocity of ice crystals is also a strong function of their shape and size characteristics. All this is considered and described in detail in (Bühl et al., 2019). Regarding the assumption on crystal shapes (plates, needles, complex forms), we make use of a variety of laboratory studies made under ambient temperature conditions (Fukuta and Takahashi, 1999; Furukawa and Wettlaufer, 2007; Myagkov et al., 2016). All in all, the terminal velocity spectrum of falling ice crystals can be retrieved with an uncertainty of about 30–50%. From the vertical velocity information particle size and the ice-crystal size distribution is retrieved.

In the next step, ICNC is estimated by comparing simulations of the ice crystal light-extinction coefficient and of the radar reflectivity, in which ICNC is now left as the only input parameter, with the measured extinction coefficients and radar reflectivity values. ICNC is varied in the simulations until the measured extinction and reflectivity values are matched. The uncertainty in the ICNC estimates would be in the order of 50% in the case of perfectly calibrated lidar and radar systems. However, realistic radar reflectivity uncertainties lead to ICNC uncertainties of a factor of 3. Such large uncertainties are still acceptable in our closure studies as we show in Sect. 5.

### 4.2.3 Observations of characteristic ice crystal terminal fall velocity (Doppler lidar) and extinction coefficient (backscatter lidar)

During the BACCHUS Cyprus-2015 campaign, we ran a Polly together with a vertically pointing Doppler lidar. A cloud radar was not available. Under these condition the entire retrieval is usually much more uncertain because of the missing Doppler spectrum information and the backscatter wavelength dependence (532 nm vs 8.5 mm). However, for the selected two case studies of the Cyprus-2015 campaign discussed in Sects. 5.2 and 5.3, stable, temporally constant ice sedimentation conditions were observed so that a good retrieval of the terminal velocity and estimation of the mean ice crystal size within an uncertainty of 30-50% was possible. ICNC is again derived from comparison of observations with respective simulations of the light extinction coefficient from lidar with ICNC as input and for an assumed size distribution adjusted to the devired mean ice crystal size. The overall uncertainty factor of 3 was also assumed in this case of ICNC estimation (applied in Sects. 5.2 and 5.3) .

At the end of Sect. 4, it is noteworthy to mention that it is practically impossible to measure, retrieve, or estimate INPC with an overall uncertainty range of less than one order of magnitude (factor 3) (DeMott et al., 2017). This holds also for many retrieval techniques and in situ measurement approaches in the case of ICNC. But, as will be shown in the next section, these uncertainties still allow us to study aerosol-cloud interaction in detail, to quantify the aerosol impact on cloud properties (such as ICNC), and to evaluate and rate the closure study as good or acceptable, or even that we failed to achieve closure between INPC and ICNC.

## 5 Field observations and closure studies

We begin with an overview of desert dust conditions over Cyprus in the spring of 2015. Figure 4 shows the frequency of occurrence of mineral dust over Nicosia during the BACCHUS Cyprus-2015 campaign. Almost every day traces of dust were visible in the lidar measurements. Dust layers were often present at heights above 5 km. Stratiform altocumulus and cirrus cloud frequently developed in these dust layers. Similar conditions were observed during the spring seasons 2016-2018 (not shown). In March-April, the rainy winter season ends and coincides with the period of frequent advection of warm and dusty air from Africa. Favorable conditions for heterogeneous ice nucleation are given during this season of the year. Examples of cloud fields evolving in Saharan dust layers during the Cyprus-2015 campaign are shown in Fig. 5. The observation of the cirrus layer in the evening of 17 March 2015 (Fig. 5e and f) and of the altocumulus layer on 8 March 2015 (Fig. 5a and b, at heights of 5-6 km after 21 UTC) will be discussed in detail in Sect. 5.2 and 5.3.

Before the presentation of the closure results, we compare the lidar-derived INPC estimates (over Nicosia) with respective in situ INPC observations taken at Agia Marina Xyliatou (see Fig. 1 and station description in Sect. 3.1). This comparison is shown in Fig. 6 and can be regarded as a good opportunity to check the reliability of the lidar INPC retrieval method. The Horizontal Ice Nucleation Chamber (HINC) (Lacher et al., 2017) of ETH (Eidgenössische Technische Hochschule), Zurich, was used for the in situ measurements performed during 20 days in March 2015 during daytime hours (at $-30°C$ and $RH_w = 104\%$, immersion/condensation freezing mode). Ten nighttime lidar observations (30-60-minute mean values, for the height layer

from 450–550 m a.s.l.) are shown for comparison. We used the immersion freezing parameterizations D15 and D10(d) (see Table 1) for the dust fraction and D10 also for the non-dust particles as described in Sect. 4.1. The lidar-derived INPC values were computed for $-30°$C. We applied the D10(d) INP parameterization in addition to the D15 parameterization because Marinou et al. (2019) found during the Cyprus-2016 campaign that INPC values obtained with the D10 parameterization were in better agreement with UAV-based in situ INPC observations than respective INPC estimates obtained with the D15 INP parameterization. The latter INPC values were a factor 2-50 higher than the in situ measured ones (at aerosol conditions described by $n_{250,d}$ of 10-30 L$^{-1}$ and for temperatures from $-20$ to $-30°$C) (Marinou et al., 2019).

According to Fig. 6, a reasonable agreement between in situ measured and lidar-estimated INPC values is found, keeping the uncertainties in the measurement and retrieval methods into account. We compare in situ measurements at rural background conditions in the Troodos mountains at Agia Marina Xyliatou with lidar observations in a lofted dust layer 220 m above ground over Nicosia. On average, the lidar-derived INPC values were higher. This is in agreement with the finding of Schrod et al. (2017). They found that the UAV-based in situ measured INPC values (for heights between 200 and 2000 m above ground) were, on average, an order of magnitude higher than the INPC values measured at the surface of the Agia Marina Xyliatou field site during the Cyprus-2016 campaign.

In the following Sects. 5.1-5.3, we present and discuss three cases of cloud evolution with focus on the ICNC vs INPC relationship. These three closure experiments cover the deposition nucleation as well as the immersion freezing regime. In the framework of these closure studies, we further check the quality and applicability of the different INPC parameterization schemes listed in Table 1. All clouds developed in pronounced Saharan dust layers. Case 1 (10 April 2017) deals with a rather thin and shallow cirrus layer that began to form at 8 km height at $-35°$ to $-36°$C in a pure Saharan dust layer (not mixed with pollution). Deposition nucleation is the prevalent ice nucleation mode at these temperatures. In Sect 5.2, we then discuss a long-lasting cirrus event (case 2, 17 March 2015) observed in the upper part of the troposphere. Cloud top temperatures were $-55°$ to $-57°$C. Homogeneous as well as heterogeneous freezing processes can take place at these cold conditions in dusty air. Finally in Sect. 5.3 (case 3, 8 March 2015), immersion freezing in an altocumulus layer developing in polluted Saharan dust at heights around 5.5–6 km and temperatures of $-19°$ to $-22°$C is discussed. The altocumulus developed in a mixture of desert dust and continental pollution particles.

## 5.1 The 10-April-2017 case study: evolution of a very thin ice cloud layer in Saharan dust

Case 1 is shown in Fig. 7 and was observed in the early morning of 10 April 2017 during the CyCARE campaign. Ideal observational conditions for an in-depth INPC-vs-ICNC closure study were given. The full LACROS equipment was running and a radiosonde was launched at 5:50 UTC (8:50 local summer time) and reached the heights around 8 km just a few minutes before ice formation started. INPC values at cloud level could be estimated in the cloud-free dusty air right before first cirrus traces crossed the lidar field site. A thin ice cloud embedded in Saharan dust developed at 8-8.2 km height at 6:30 UTC. Temperatures were $-35°$C to $-36°$C at these heights. The particle depolarization ratio indicated ice crystal backscattering from the beginning of cirrus formation (the absence of any liquid phase) so that deposition nucleation or PCF (because this

would also occur without a bulk liquid phase being detected) was obviously exclusively responsible for ice crystal nucleation. The available INPs thus determine the maximum number of observable ice crystals.

The backward trajectories in Fig. 8 suggest that the air masses that reached Limassol at 8 km height crossed the Sahara at comparably high altitudes of 4.5-5 km height. Marinou et al. (2017) showed that dust layers over northern Africa (Algeria, Libya) reach to 6 km throughout the year, except during the main winter months (December-January) so that significant dust uptake is always possible during spring months as long as the trajectories are below 6 km over northern Africa.

In Fig. 9, the main closure results are summarized. A pronounced dust layer was present from 7.2 to 9.3 km height at times before the cirrus developed (6:10-6:25 UTC) and showed 532 nm particle extinction coefficients of about 25 $Mm^{-1}$ at 8 km height (see Fig. 9a). The particle depolarization ratio (not shown) was around 0.3 in the dust layer. This means that dust particles dominated and contribution from non-dust aerosol particles to the overall particle lidar backscatter return signals was negligible. The radiosonde $RH_w$ indicated a moist layer which coincided with the dust layer. The air in the middle troposphere (at 5 km height) and at cirrus level (10 km) was dust free according to the trajectory analysis in Fig. 8 and our lidar observations.

The dust extinction coefficients were then converted into number concentrations of large particles, $n_{250,d}$, and surface area concentration $s_d$. At cloud level, the peak $n_{250,d}$ and $s_d$ values were close to 5500 $L^{-1}$ and around $85 \times 10^{-12} m^2 cm^{-3}$, respectively (Fig. 9b). By applying the INPC parameterization schemes U17-I(d), U17-D(d), and D15 (see Table 1) with $n_{250,d}$, $s_d$, the corresponding GDAS1 temperature profiles as input, and assumed values of the ice supersaturation $S_i$ from 1.05–1.35 in the U17-D(d) computations, we obtain the INPC profiles shown in Fig. 9c. The immersion-freezing INPC profiles (although not relevant here) are included in the figure to visualize the high INP numbers of $>1000 L^{-1}$ if immersion freezing would have been the dominant ice-nucleation mode. As we will discuss below, such high numbers are in strong contradiction with the estimated ICNC that were clearly $<100 L^{-1}$ as shown in Fig. 9c as well.

In the second part of the closure experiment, we estimated ICNC in the thin ice cloud. In Fig. 10, the observations with the vertically pointing Doppler lidar and Doppler radar are shown. The Doppler lidar detected first ice crystals around 6:30 UTC (see Fig. 10d) in agreement with the more powerful Polly observations (with higher signal-to-noise ratios) shown in Fig. 7. First significantly enhanced attenuated backscatter values were recorded about 5-7 minutes later (see Fig. 10a). Vertical velocities were positive (upward motion) and in the range of 0–0.5 $m\,s^{-1}$ around 6:30 UTC as the Doppler lidar measurements revealed and triggered first ice nucleation events. A pronounced region with positive vertical velocities, indicating a well organized updraft, then occurred between 6:50 and 6:57 UTC. This lifting intensified ice crystal nucleation and growth. Afterwards a pronounced area with descending particles became visible, especially in the Doppler radar observations (Fig. 10c).

This time period from 6:57–7:05 UTC was used to estimate ICNC following the method described in Sect. 4.2.2. The analysis yielded a mean crystal terminal velocity of 0.29 $m\,s^{-1}$ and an ICNC of 13 $L^{-1}$ (as given in Figs. 9a and c). The ice crystal extinction coefficient peaked at 710 $Mm^{-1}$ (not shown, in the case of the blue cirrus profile in Fig. 9a). The cirrus optical thickness was 0.02-0.05 (6:35-6:48 UTC) and then 0.1-0.2 (6:48-7:07 UTC). An overview of all ICNC and INPC analysis products for 10 April 2017 is given in Table 2. Taking an uncertainty of a factor of 3 into account, ICNC was in the range of 4–39 $L^{-1}$.

The 35.5 GHz cloud radar (8.5 mm wavelength) provided useful reflectivity and Doppler information, caused by growing ice crystals, not before 6:52 UTC. During the initial phase (6:30–6:50 UTC) the cirrus remained invisible for the radar (see light blue profile in Fig. 9a with a peak ice extinction coefficient of 115-120 $Mm^{-1}$). Our simulations with the cirrus information obtained from the Polly and Doppler lidar observations and by assuming a radar reflectivity value (clearly below the detection limit) reveal that ICNC was probably in the range of 2–10 $L^{-1}$ before 6:50 UTC. Here, we used the approach described in Sect. 4.2.1. However, we cannot fully exclude that a rather narrow ice crystal spectrum with crystal sizes of 10-20 $\mu$m was present at the beginning of the cirrus formation and ICNC was as high as 100-200 $L^{-1}$. However, such high numbers of ICNC would be in contradiction with the lidar-based INPC estimates of $< 50$ $L^{-1}$ as discussed above and shown in Fig. 9c.

As shown in Fig. 9a (light blue curve, 6:35-6:42 UTC) the thin cirrus layer developed in the center of the dust layer. No indication for strong lifting of the Saharan air mass and rapid growth of the nucleated ice crystals and subsequent evolution of fall streaks were visible within the first 20 minutes (6:30-6:50 UTC) of the evolving ice layer (see Fig. 7). It seems that the ice supersaturation level $S_i$ for the initiation of heterogeneous ice nuclation was slowly reached so that ice crystals formed, but then $S_i$ decreased again towards ice saturation ($S_i = 1.0$) because of water vapor deposition on the nucleated ice crystals and, as a consequence, strong growth of ice crystals and the formation of a virga zone were not possible. Virga were absent during the first 20 minutes of cloud evolution.

Regarding the required ice supersaturation conditions, we assume that large-scale lifting and advection of moist air was responsible for the evolution of the cirrus layer at 8 km height. The radiosonde measured a relative humidity over water ($RH_w$) of 63% at 8.2 km height at about 6:30 UTC, obviously in clear air, i.e., before cirrus formation started. As can be seen in Fig. 9c, closure between ICNC of 13 $L^{-1}$ (or from 4-39 $L^{-1}$) in the developing cirrus layer and INPC at cloud top of 8.4-8.5 km is obtained for ice supersaturation values around 1.2-1.25 (or $RH_i$ of 120–125%) which corresponds to $RH_w$ of about 90–95% at temperatures of $-35°C$ to $-36°C$. It is interesting to note that Roberts and Hallett (1968) and Schaller and Fukuta (1979) found in laboratory studies more than 40 years ago that deposition nucleation occurs when $S_i$ exceeds 1.2. This finding is in good agreement with our observation and the applied U17-D INP parameterization. Nevertheless, according to Fig. 9c deposition nucleation can occur at any $RH_i > 100\%$ and $S_i > 1.0$ as already found earlier (Mangold et al., 2005; Kanji and Abbatt, 2006).

In this context, it should be mentioned (in order to avoid confusion about the apparently high INPC numbers outside the clouds) that the presented INPC profiles are given for fixed, height-independent ice supersaturation levels. In reality, these high supersaturation values typically only hold for those parts of the cloud layers (e.g., from 8.2-8.5 km height in Fig. 9) in which ice nucleation takes place. Outside the clouds (and below the ice virga zone) ice subsaturation ($S_i < 1.0$) is given so that INPC=0.

As a final remark, it should also be emphasized that the unknown vertical motions have a sensitive impact on the actually active INPs and thus on the success of the INPC vs ICNC closure study. Even weak short-lived updrafts associated with cooling of the air by 1-2°C and a respective increase in ice supersaturation can easily increase INPC by a factor of 2-3 (for temporally constant dust conditions).

The successful closure between INPC and ICNC as given in Fig. 9c for $S_i = 1.25$ and ICNC of 13 $L^{-1}$ suggests also that the U17-D INP parameterization scheme is useful and allows a trustworthy estimation of dust INPC from measured profiles of $s_d$. This finding is corroborated by the study of Marinou et al. (2019) who also observed good agreement between lidar-based

INPC obtained by means of U17-D(d) and respective UAV-based in situ observations of deposition-nucleation INPC during the Cyprus-2016 campaign.

## 5.2 The 17-March-2015 case study: long-lasting cirrus evolution in Saharan dust

The next closure study deals with cirrus formation in the upper troposphere. This case is shown in Fig. 5e and f and was
measured during the Cyprus-2015 campaign (see Sect. 3.1). The ice cloud developed in the upper part of a Saharan dust layer. According to the backward trajectory analysis in Fig. 11, Saharan dust was lifted from heights close to the surface up to the upper troposphere over Africa two days before arriving over Nicosia. We observed the final phase of a complex and long-lasting cloud evolution event which started on 15 March 2015 over northern Africa. The particle linear depolarization ratio measured with lidar during cloud-free periods was again around 0.3 throughout the dust layer and indicated pure desert dust conditions.
The trajectory analysis in Fig. 11 further reveals that the boundary layer aerosol was advected from Turkey and crossed Europe days before, whereas the dust layer from 2-6 km (see Fig. 5f) contained a mixture of dust from Iraq and continental pollution aerosol. Such complex mixtures and layering of aerosols frequently occur over Cyprus.

This cloud case deserves a detailed discussion of the entire life cycle. A dust-infused baroclinic storm (DIBS, a synoptic-scale dynamics driven storm) (Fromm et al., 2016; Caffrey et al., 2018) associated with vigorous convective motions within
cloud towers were responsible for the uplift of large amounts of dust into the upper troposphere over the Sahara, observed with lidar later on over Cyprus. According to Fig. 12a several fronts crossed the desert areas of northwestern Africa from the west in the morning and afternoon of 15 March 2015. The DIBS developed over the desert areas over eastern Algeria from 18:00-24:00 UTC on 15 March 2015 (Fig. 12b). Dust lifting obviously occurred also before and after the main cloud convection period according to HYSPLIT backward trajectories in combination with our continuous lidar observations. The convective
cluster with a large anvil cirrus on top moved northeastward and started to dissolve over the central Mediterranean Sea on 16 March 2015, 3:00-6:00 UTC (Fig. 12c). The large anvil cirrus shield covering an area of approximately 600 km × 1200 km over the central and eastern Mediterranean is visible in Fig. 12d. The anvil cirrus deck weakened and became then transformed into an active cirrus uncinus field which persistently generated new ice crystals and virga. This cirrus field extended from Crete to Tajikstan (as confirmed by our Polly lidar observations at Dushanbe, Tajikstan, in the afternoon of 17 March 2015), and
thus over a distance of more than 3500 km (Fig. 12g). These cirrus uncinus structures were visible over Nicosia from about 7:00 UTC in 17 March 2015 to 15:00 UTC on 18 March 2015 as will be discussed below.

Cirrus uncinus belongs to the synoptic cirrus category (Sassen, 2002). These ice clouds form in situ in the upper troposphere in response to a variety of weather disturbances. They typically form from the top down (i.e, ice crystal nucleation occurs at cloud top and subsequent sedimentation of ice crystals lead to an extended virga zone). The almost constant (non-descending)
cirrus top height, observed over the whole day of 17 March 2015 with lidar, indicated an active ice cloud that continuously produced new ice crystals. Usually dissoloving anvil cirrus fields descend with time (Strandgren, 2018). However, these clouds were visible in the satellite images until 18 March, about three days after the formation of the DIBS-related cloud complex over Algeria and did not descend according to our lidar observations. These long-living cirrus features were probably the result of favorable meteorological conditions (high humidity, permanent occurrence of vertical motions) (Feng et al., 2012)

in combination with the high dust load in the upper tropopshere serving as an almost unlimited reservoir of INPs. Feng et al. (2012) reported that typical lifetimes of midlatitudinal anvil cirrus systems are <3 h in 50% of observed cases, the majority shows lifetimes <15 h. Here, we have an overall cirrus life time of 2-2.5 days, and thus all in all of 48-60 hours. Note, that reduced cirrus lifetimes are usually assumed when heterogeneous (instead of homogeneous) ice nucleation dominates (see,

e.g., Storelvmo et al., 2018; Gruber et al., 2019). This may indeed be the case in controlled seeding experiments or, more generally, in cases with a limited, depletable source of INPs.

Following the terminology of Kuebbeler et al. (2014), our closure study in Fig. 13 describes ice nucleation at pre-existing ice conditions. Strong supersaturation over ice is no longer possible at pre-existing ice conditions, i.e., in a fully developed cirrus system, so that homogeneous freezing can be excluded in our closure analysis. Homogeneous freezing needs stable conditions

regarding lifting of air parcels over a long time period so that $S_i$ steadily rises and can finally reach values of 1.4-1.7. Number concentrations of nucleated ice crystals may be of the order of 1000 $L^{-1}$ in the case of homogeneous freezing (Kärcher et al., 2006; Kuebbeler et al., 2014). However, at pre-existing cirrus conditions, positive and negative vertical motions change frequently within the cirrus layer as a result of radiative cooling at cloud top, turbulence and entrainment processes and under the permanent influence of short-term gravity wave activity so that it is very likely that $S_i$ remains below 1.2 at all. At these

conditions, we assume that new crystals form heterogeneously in updraft parcels, grow quickly by water vapor deposition, and start falling. These ice crystals leave the upper part of the cloud and are replaced by ice crystals freshly nucleated in the dusty air. At theses steady-state conditions, the number concentration of ice crystals is controlled by the number of available INPs (Kärcher et al., 2006; Kuebbeler et al., 2014) and ice nucleation is exclusively caused by deposition nucleation (in Fig. 13 at $-55$ to $-57°$C).

The main closure results are summarized in Fig. 13. According to the dust extinction coefficient of 10–20 $Mm^{-1}$ in the upper part of the pronounced dust layer from 6-10 km height (red profile in Fig. 13a), particle number concentrations $n_{250,d}$ of large particles were mostly between 2000 and 3000 $L^{-1}$ and dust particle surface area concentration $s_d$ between 40 and $50×10^{-12}m^2cm^{-3}$ (Fig. 13b). The aerosol observations were performed after the passage of the day-long cirrus field, from 23:05-23:00 UTC. The higher cirrus layer (deep blue profile in Fig. 13a, measured from 20:00-20:40 UTC) started to develop

at heights from 10–11 km and obviously depleted the dust layer of INPs in this height range. The second cirrus with top height at 10 km was observed between 22:10–22:19 UTC (light blue profile in Fig. 13a), about 45 minutes before the cirrus-free aerosol observations were conducted. This ice layer started to form just at the top of the pronounced dust layer, and thus in a less dust-depleted environment.

The INPC height profiles for different reasonable $S_i$ values shown in Fig. 13c are derived from the lidar observations by

means of the U17-D(d) parameterization (Table 1). As already mentioned in the preceding section, it should be emphasized here again that the dust conditions at 23:05–23:30 UTC can only provide a realistic range of INP numbers. The true ones, responsible for ice nucleations from 20:00-20:40 UTC (first cirrus layer) and 22:15–22:19 UTC (second cloud layer) remain unknown because of unknown vertical motions, cooling rates, and variability in the dust concentration. We bear in mind this caveat in the interpretation of the results of the INPC-vs-ICNC closure experiments. Also the unknown ICNC reduction by,

e.g., ice crystal collision and aggregation processes need to be kept in consideration in cases with cirrus vertical extent of 1.5-2 km and cirrus optical thickness of 0.25 (22:15-22:19 UTC) to 0.75 (20:00-20:40 UTC).

Regarding the estimation of the ICNC values only the vertically pointing Doppler lidar and the backscatter lidar Polly were available during the Cyprus-2015 campaign. From the continuous vertical-velocity observations with the Doppler lidar over hours, a clear picture of the mean terminal velocity of falling ice crystals in the lower part of the cirrus (see sketch in Fig. 3a) could be obtained. By using this terminal-velocity information together with the peak values of the ice extinction coefficient of 760 $Mm^{-1}$ (not shown in the case of the dark blue curve in Fig. 13a) and 310 $Mm^{-1}$ (also not shown), both occurring at 9.1-9.2 km height, the analysis (see Sect. 4.2.3) revealed ICNC values of 16 $L^{-1}$ and 5 $L^{-1}$ for the two analyzed cirrus segments. These ICNC values plus uncertainty margin of a factor of 3 are indicated as vertical lines in Fig. 13c. The results of the ICNC and INPC data analysis for this case are also presented in Table 2.

As can be seen, an acceptable agreement of INPC with ICNC is obtained for $S_i$ around 1.1. Keeping an underestimation of INPC in updrafts into consideration, even lower ice supersaturation values of 1.05-1.07 were probably sufficient to maintain the observed cirrus conditions and to obtain agreement between the numbers of ICNC and INPC. A reduction of ICNC by, e.g., aggregation effects, may have occurred as well so that our ICNC value (representing the cirrus conditions in the lower half of the cloud layer) was too low (with respect to the ICNC at cloud top) by a factor of 5. Agreement between ICNC and INPC is then only reached when $S_i$ is in the range of 1.15. All this is in good agreement with modeling results presented by Kuebbeler et al. (2014) for pre-existing cirrus conditions. The consistent picture of the cirrus evolution in terms of INPC and ICNC again corroborates the usefulness of the U17-D parameterization to predict dust INPC even at temperatures below $-55°C$.

Fig. 14 provides an overview of the entire cloud evolution from 17 March, 18:00 UTC, to 18 March, 24:00 UTC. As mentioned, the cirrus layer was monitored over the whole day on 17 March (from 7:00–22:45 UTC). Another cirrus field started to cross the lidar site at midnight (17 March 2015, 23:45 UTC) and the evolution of this cirrus system was controlled by a low pressure system over Turkey that moved slowly eastward. With time, the dusty, humid African air was replaced by dust-free and drier air from the North. As a consequence the cirrus deck apparently descended (embedded in Saharan dust) from 00:00 UTC to about 15:00 UTC from 10 to 3-4 km height and transformed into extended fields of mixed-phase altocumulus layers during the second half of 18 March 2015. A new dust outbreak advecting dusty air from Africa reached Cyprus at about 17:00 UTC on 18 March 2015. The composite shows the full advantage of continuously running remote sensing instruments. Such coherent observations of cloud processes are not possible by any other means. The transition from the deposition-nucleation regime with cloud top temperatures $< -35°C$ to the immersion freezing regime with cloud top temperatures $> -30°C$ was covered by the lidar observations. Seeder-feeder effects (Rutledge and Hobbs, 1983; Fleishauer et al., 2002; Ansmann et al., 2009) with pronounced virga formation below the lowest liquid water cloud layers around 5 km height are visible in the figure, especially from 11:00 to 18:00 UTC, before the new dusty air mass arrived. In this air, altocumulus developed and immersion freezing processes were responsible for the observed ice formation.

INPC and ICNC values are given for several orange and blue boxes (indicating signal averaging periods and data analysis height ranges) in Fig. 14. The numbers for 17 March 2015 (cirrus in the upper troposphere) were already discussed above. Good to acceptable agreement between the estimated INPC and ICNC values was found in this case with deposition nucleation. The

agreement is less good for the altocumulus layer observed in the evening of 18 March. For cloud-free periods INPC values were determined and ranged from 0.1 to about 5 L$^{-1}$. For several cloud segments with cloud top height between 5 to 6.5 km height, quite different ICNC values were derived, ranging from 0.2–2 L$^{-1}$ to 4-36 L$^{-1}$. Besides the uncertainties in the remote-sensing-based INPC estimation (already considered in the shown INPC numbers in Fig. 14) ice break-up processes

and crystal collision events associated with crystal aggregation formation which leads to an increase and reduction of ICNC, respectively, must be kept in mind in the closure studies at these relative high ice formation temperatures ranging from about $-15$ to $-25°$C. These problems mostly associated with high ICNC-to-INPC ratios were already reported by Auer et al. (1969) and Hobbs (1969) (see Sect. 2).

## 5.3    The 8-March-2015 case study: evolution of a mixed-phase altocumulus layer in polluted Saharan dust

In the first two closure studies, we discussed ice nucleation processes at relatively low temperatures of $-35°$C to $-57°$C and found that deposition nucleation explained the ICNC observed. The third case now deals with heterogeneous ice nucleation within a stratiform mixed-phase altocumulus layer with cloud top temperatures of $-20$ to $-23°$C. As mentioned, immersion freezing is the prevalent ice nucleation mode at these temperatures.

The altocumulus developed over Nicosia in desert dust at heights around 6 km in the late evening of 8 March 2015 (after

21:00 UTC). The case is shown in Fig. 5a and b. According to the backward trajectories in Fig. 15, the dust above 4 km height originated from the Sahara, whereas the dust layer between 1 and 3.5 km was aged dust from the Middle East mixed with eastern European aerosol pollution.

Fig. 16 presents the basic lidar aerosol products for the time period just before the cloud layers appeared (22:00-22:20 UTC). The dust from northern Africa was contaminated by anthropogenic fine-mode aerosol because the measured particles depo-

larization ratio was significantly below the pure dust value of 0.3. Small particles cause depolarization ratios of less than 0.05 so that depolarization ratio values around 0.2 indicate a significant contribution of continental fine-mode particles (indicated by the index c) to total particle backscattering and extinction as shown in Fig. 16. After separation of the dust and non-dust continental extinction coefficients $\sigma_d$ and $\sigma_c$ as described in detail by Mamouri and Ansmann (2014), the particle extinction profiles were used as input in the retrieval of the particle number concentrations $n_{250,d}$ and $n_{250,c}$ and of the particle surface

area concentrations $s_d$ and $s_c$ in Fig. 17b and in the INPC estimation in Fig. 17c as outlined in Sect. 4.1. In Fig. 17b, $n_{250,d}$ values were as high as 1800 L$^{-1}$ at 5.8 km height (cloud level), $n_{250,c}$ close to 500 L$^{-1}$ and surface area concentrations $s_d$ and $s_c$ reached values of $30\times10^{-12}$m$^2$cm$^{-3}$ and $20\times10^{-12}$m$^2$cm$^{-3}$, respectively.

For the non-dust continental aerosol fraction we then applied the U17-I(c) INP parameterization (Table 1), and for the dust fraction we used the D10(d), D15 and U17-I(d) INP parameterizations. We show all obtained INPC profiles in Fig. 17c to give

an impression of the uncertainty range. The dust INPC estimates obtained with D15 and U17-I(d) differ by almost a factor 50 (see also Table 1, at 5.8 km height and $-20°$C). The D10(d) solutions are close to the retrieved ICNC values in Table 1 and may be thus more reliable than the other solutions as found also by Boose et al. (2016) and Marinou et al. (2019). In both studies, it was found that the D10(d) parameterization (with in situ measured $n_{250,d}$ as input) yielded better agreement with in situ measured INPC values at Tenerife and Cyprus than the use of the D15 approach. Further discussion on uncertainties in

INPC measurement methods, predictions, and parameterizations can be found in Phillips et al. (2013); Price et al. (2018), and DeMott et al. (2017).

The reason for the large difference between the D15 and U71-I(d) INPC values of $0.13 \, \text{L}^{-1}$ and $6.3 \, \text{L}^{-1}$, respectively remain unclear. DeMott et al. (2015) introduced a calibration factor cf = 3.0 to obtain good agreement between the D15 INPC values
and the INPC values when using the parameterization of Niemand et al. (2012), denoted here as N12 parameterization. The U17-I(d) INPC numbers are a constant factor of 1.64 higher than the respective N12 INPC numbers (Ullrich et al., 2017) so that a difference of a factor of about 5 ($3 \times 1.64$) between the D15 solutions (with cf = 1.0 in our closure study) and the U17-I(d) results can explain a part of the difference. However even after multiplication of the D15 INPC by a factor of 5, yielding $0.65 \, \text{L}^{-1}$, an order of magnitude remains between the U17-I(d) and the increased D15 INPC values. This may be partly related
to the fact that the D15 parameterization (INPC versus temperature) is different for low and high dust concentration levels. The D15 and N12 INPC comparisons discussed in DeMott et al. (2015) were performed at dust outbreak conditions over Cabo Verde with dust extinction coefficients around $100 \, \text{Mm}^{-1}$. Our U17-I(d)-vs-D15 INPC comparison is performed for dust conditions with dust extinction coefficients of $10 \, \text{Mm}^{-1}$ at 5.8 km height. For such low dust concentrations, a much larger calibration factor of cf > 6.0 is needed in the D15 INPC retrieval to match the N12 INPC numbers (DeMott et al., 2015). It is
also possible that the INPC reservoir comprises even smaller particles, e.g., particles with radius >100-150 nm. This would lead to increase of a factor of 3-4 of the D15 INPC numbers compared to the INPC values obtained with $n_{250}$.. Vice versa, the surface-area concentration used in the U17-I(d) INPC compution may be too large. Only the surface area for those particles (with radius >100 nm) should be considered that serve as cloud condensation nuclei (Ansmann et al., 2019). The respective surface area concentration is a factor of 1.5-2 lower than the total particle surface area. It remains to be mentioned that a
similar deviation of the U17-I(d) from the D15 INPC numbers as shown in Fig. 17c was obtained in the studies of Marinou et al. (2019).

Regarding the estimation of the ICNC values, we followed the strategy illustrated in Fig. 3b. The first altocumulus layer (22:29-22:31 UTC) started to form at 5.6 to 6.1 km height (dark blue profile in Fig. 17a) in the center of the upper dust layer (peaking at 5.8 km). Peak cloud extinction values (not shown) in the main altocumulus layer of $5000 \, \text{Mm}^{-1}$ (in the case of
the light blue curve) to $8000 \, \text{Mm}^{-1}$ (in the case of the blue curve) occurring at heights from 6-6.2 km height, i.e., at the cloud top are typical for cloud layers dominated by liquid-droplet backscattering. The depolarization ratio also dropped to low values typical for spherical droplets in this droplet-dominated layer. Virga developed (see Fig. 5a and b) and caused ice particle extinction coefficients of $100-150 \, \text{Mm}^{-1}$ below 5.6 km height (see Fig. 17a). Later, the second cloud layer (22:43-22:53 UTC) was found roughly at 250 m higher so that the cloud top temperature decreased by 2°C which means that the
INPC values increased by a factor of roughly 2-4 assuming that the dust particle concentrations remained almost constant. As a consequence of the increased availability of INPs, also more crystals nucleated and consequently ice extinction coefficients in the virga zone increased to $150-180 \, \text{Mm}^{-1}$ in Fig. 17a (ice virga zone from 4.7-5.7 km height).

For completeness it should be mentioned that the primary lidar cloud parameter is the cloud backscatter coefficient. The liquid-water extinction coefficients are obtained by multiplying the strong cloud backscatter coefficients (in the cloud top region
of the altocumulus layer) with the droplet lidar ratio of 18 sr (O'Connor et al., 2004). The ice crystal extinction coefficients in

the ice virga zone are obtained as described in the sections before by using a typical cirrus lidar ratio of 32 sr (Seifert et al., 2007; Giannakaki et al., 2007) in the conversion of the ice crystal backscatter values into the extinction coefficients. Note also, that the backscatter coefficient profiles are determined by means of the Raman lidar method (Ansmann et al., 1992) in which no lidar ratio profile is needed as input, in contrast to the widely and commonly used Fernald backscatter retrieval technique (Fernald, 1984) so that the obtained backscatter profiles are very accurate and not corrupted by strong height variations of the lidar ratio (from the dust layer below the cloud dust over ice virga zone to the liquid-water layer at cloud top).

The Doppler lidar measurements of the fall speed of the ice crystals and the ice extinction values of 120 and 180 $Mm^{-1}$ in the virga zone were then used to estimate the ICNC numbers in Fig. 17a for the two cloud cases following the methodology described in Sect. 4.2.3. By considering an uncertainty of a factor of 3, ICNC was between 0.3 and 7.5 $L^{-1}$.

As mentioned, the comparison of estimated ICNC and INPC values is difficult in view of the large differences between the different INPC profiles. However, if we use the ICNC values as a guide line for the true INPC levels then the D10(d) INPC values (ignoring a small non-dust INPC contribution) is closest to the ICNC values at 5.8 km height. The respective D15 and U17-I(d) profiles in Fig. 17c may be used as minimum and maximum boundaries of the INPC uncertainty range, as it was already suggested by Marinou et al. (2019).

The cloud top height increased by 250 m from cloud segment 1 to cloud segment 2 in Fig. 17a. The respective decrease in temperature leads to an increase in INPC by a factor of 2-3. This lifting of cloud and dust parcels is not included in the INPC profiles (estimated from the aerosol conditions before the cloud field arrived and before the lifting took place). All in all, we rate this closure experiment also as successful. As already discussed above, a better agreement cannot be expected. With increasing cloud top temperature the probability of secondary ice formation and thus of strong changes in the ICNC numbers increases. Furthermore, ice crystals nucleated via immersion freezing can grow very fast in the (almost) unlimited liquid-water reservoir and then probably create ice crystal clusters with complex shape features which may intensify ice-break-up as well as collision processes.

## 6 Summary and conclusions

(1) For the first time closure studies of the relationship between ice-nucleating particle concentration INPC and ice crystal number concentrations ICNC in altocumulus and cirrus layers, solely based on ground-based active remote sensing, has been conducted. Three closure experiments (proof-of-concept studies) covering the deposition nucleation and immersion freezing regimes were presented. All clouds developed in pronounced Saharan dust layers. We rated the closure study as successful if the estimated INPC and ICNC values agreed within an order of magnitude (i.e., if the INPC and ICNC uncertainty ranges of one order of magnitude widely overlapped). All three closure experimente were classified as successful.

A long-lasting, complex evolution of a DIBS Fromm et al. (2016)) associated with the formation of an usual large anvil cirrus shield and the development of rather large cirrus uncinus field extending from the Eastern Mediterranean to Central Asia over more than 3000 km was investigated. The unusual characteristics (long life time, large coverage with anvil and synoptic cirrus fields) lead to the conclusion that the presence of dust (i.e., of an unlimited reservoir of INPC) in the upper troposphere

sensitively influenced the cloud evolution processes and extended the cloud lifetime. Such cloud evolution cases with largely extended cloud life time and increased regional coverage may be counted as contribution to anthropogenic climate forcing if it can be shown that the occurrence frequency and strength of these MCSs, able to lift large amount of natural INPs into the upper troposphere, increased during the last decades in a warming climate caused by man-made activity.

(2) Our remote-sensing-based closure experiments demonstrated that dust can trigger significant ice formation in the middle and upper troposphere via the heterogeneous ice nucleation. The homogeneous freezing path way was not needed to explain the evolution of the cirrus uncinus fields at $-55$ to $-57°C$ (case 2 of our closure studies).

(3) A new lidar-radar-based methodology based on new ICNC (Sourdeval et al., 2018; Mitchell et al., 2018; Bühl et al., 2019) and INPC retrieval techniques (Mamouri and Ansmann, 2015, 2016; Marinou et al., 2019) is now available to investigate the role and impact of aerosol particles on ice formation in atmospheric clouds and on subsequent precipitation processes. Especially, lidar profiling of INPC can be regarded as an important step forward in the research field dealing with aerosol-cloud interaction via ice formation. With view on a potential consideration and implementation of complex aerosol-cloud interaction in numerical weather forecast models and assimilation of measured INPC profiles into those models, there is no alternative to continuous lidar-based INPC profiling and monitoring techniques. This fact corroborates the importance of lidar and the need for further in-depth laboratory and field studies to obtain robust and trustworthy INPC parameterization schemes applicable to ground-based and spaceborne lidar measurements. Although the INPC retrieval input parameters (large particle fraction, particle surface area concentration) can be determined with lidar with an uncertainty of the order of 25-30%, the overall uncertainty of the lidar-based INPC profiling is in the range of one order of magnitude caused by the available INPC parameterization schemes.

It can be concluded from these comparisons that further efforts are needed to evaluate the reliability and usefulness of the developed INPC parameterization schemes used to predict INPC in the middle and upper troposphere (see recent publications of Welti et al. (2019) and Harrison et al. (2019)). In situ observations aboard aircraft overflying sophisticated remote sensing field sites (equipped with state-of-the-art instrumentation for detailed aerosol, cloud and meteorological measurements) are required to allow in-depth comparisons of measured, retrieved, and estimated INPC profiles up to tropopause level. Although UAVs are mostly used in atmospheric research at heights below 3 km, UAV-Balloon systems are under development that can even reach stratospheric heights and will be able to profile particle size distribution and other aerosol properties within the entire troposphere. We also need such kind of in-situ-vs-remote-sensing studies to validate our developed remote-sensing-based ICNC retrieval method. Airborne in situ observations of ice crystal size distributions, shape properties, and ICNC in ice clouds over the remote sensing facility would be desirable for intensive ICNC comparisons.

(4) As a significant contribution to climate-change research, we need to apply the INPC-vs-ICNC closure concept to space-borne lidar-radar studies to obtain an improved view on aerosol-cloud-precipitation relationships and the role of different aerosol types (marine, anthropogenic, and dust particles) on ice-formation processes on a global scale. A first attempt was shown by Marinou et al. (2019). Complementary, we need ground-based long-term field campaigns such as CyCARE in Cyprus and DACAPO-PESO (Dynamics, Aerosol, Cloud And Precipitation Observations in the Pristine Environment of the Southern Ocean) to improve cloud process understanding at very different places (climate zones) with very contrasting weather

regimes and aerosol conditions. We moved our lidar-radar LACROS equipment after the CyCARE campaign (October 2016 - March 2018) to Punta Arenas at the southern most tip of South America (long term measurements started DACAPO-PESO at the end of November 2018) to investigate aerosol cloud interaction at rather pristine conditions, with only a few episodes of siginfcant amounts of continental aerosols (smoke and dust from southern parts of South America, and long-term transport for Australia).

(5) As an outlook and to reduce the number of critical assumptions in our closure methodology as discussed in Sect. 4, we may include the next generation of powerful water vapor differential absorption lidars (DIALs) or Raman lidars to obtain temporally and vertically highly resolved water vapor and relative humidity profiles in cirrus and transparent altocumulus layers (Leblanc et al., 2012; Reichardt et al., 2012; Späth et al., 2016; Sakai et al., 2019) as well as information on liquid-water and ice-water content (Wang et al., 2004; Sakai et al., 2013; Reichardt, 2014) in future closure studies. We may also integrate radar wind profilers in our cloud studies to obtain detailed updraft and downdraft observations in the cloud top regions (Bühl et al., 2015; Radenz et al., 2018), and even lidar techniques for quantification of mineral dust concentrations within the ice clouds (Tatarov and Sugimoto, 2005; Müller et al., 2010; Tatarov et al., 2011).

## 7  Data availability

Polly lidar observations (level 0 data, measured signals) are in the PollyNET data base (http://polly.rsd.tropos.de/). LACROS observations (level 0 data) are stored in the Cloudnet data base (http://lacros.rsd.tropos.de). All the analysis products are available at TROPOS upon request (info@tropos.de). Backward trajectories analysis has been supported by air mass transport computation with the NOAA (National Oceanic and Atmospheric Administration) HYSPLIT (HYbrid Single-Particle Lagrangian Integrated Trajectory) model (HYSPLIT, 2019) using GDAS1 meteorological data (Stein et al., 2015; Rolph et al., 2017). GDAS1 data is available via the ARL webpage https://www.ready.noaa.gov/gdas1.php.

## 8  Author contributions

AA, REM, and JB prepared the manuscript. JB, PS, RE, REM, and AN took care of Cyprus-2015 and CyCARE data collections and analysis. JH was involved in the CyCARE Polly data analysis. JA, ZK, and BS performed and analyzed the HINC-based in situ INPC observations at Agia Marina Xyliatou. MV and JS were responsible for all Agia Marina Xyliatou field activities, including the UAV operation, data collection and analysis and took care of the well prepared infrastructure and logistics of the remote sensing field site at the Cyprus Institute in Nicosia during the Cyprus-2015 and Cyprus-2016 campaigns.

## 9  Competing interests

The authors declare that they have no conflict of interest.

## 10 Special issue statement

This article is part of the special issue "BACCHUS – Impact of Biogenic versus Anthropogenic emissions on Clouds and Climate: towards a Holistic UnderStanding (ACP/AMT/GMD inter-journal SI)". It is not associated with a conference.

*Acknowledgements.* The authors acknowledge funding from the European FP7 project by the European Union's Seventh Framework Program (FP7/2007-2013) collaborative project BACCHUS (grant agreement no. 603445) and the Horizon 2020 research and innovation program ACTRIS-2 Integrating Activities (H2020-INFRAIA-2014-2015, grant agreement no. 654109). We are also grateful to the National Observatory of Athens (NOA, Vassilis Amiridis) for providing the Polly lidar for the 6 week BACCHUS field campaign in 2015. The NOA Polly lidar was supported by the EU FP7-REGPOT-2012-2013-1BEYOND project (Building Capacity for a Centre of Excellence for EO-based monitoring of Natural Disasters) under grant agreement no. 316210. The CyCARE/A-LIFE data analysis has been supported by the SIROCCO project (EXCELLENCE/1216/0217) co-funded by the Republic of Cyprus and the Structural funds of the European Union for Cyprus through the Research Promotion Foundation. MV acknowledges funding from the University of Bremen and the DFG-Research Center/Cluster of Excellence: The Ocean in the Earth System-MARUM. Aerosol sources apportionment analysis has been supported by air mass transport computation with the HYSPLIT model using GDAS1 meteorological data. We acknowledge the provision of GDAS1 data by the NOAA-Air Resources Laboratory.

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

**Table 1.** INPC parameterizations used in our study and taken from the literature (reference and specific equation in this reference). Details of the lidar-based method to estimate INPC profiles can be found in Mamouri and Ansmann (2016) and Marinou et al. (2019). Input parameters in the INPC estimation are the particle number concentrations $n_{250,\mathrm{d}}$ and $n_{250,\mathrm{c}}$, the particle surface area concentrations $s_{\mathrm{d}}$ and $s_{\mathrm{c}}$, and temperature $T$. Index d and c denote dust particles and continental pollution particles (e.g., haze, smoke). The D15 INPC parameterization is used with a factor (cf in Eq. (1) in D15) of 1.0 in our study instead of 3.0 as applied in the original equation (DeMott et al., 2015). In the U17-D INP parameterization the ice supersaturation value $S_{\mathrm{i}}$ has to be set.

| INP method | Reference, equation | Nucleation mode | Aerosol type | Input |
|------------|---------------------|-----------------|--------------|-------|
| D10 | DeMott et al. (2010), Eq. (1) | Immersion | continental aerosol | $n_{250,\mathrm{c}}, T$ |
| D10(d) | DeMott et al. (2010), Eq. (1) | Immersion | desert dust | $n_{250,\mathrm{d}}, T$ |
| D15 | DeMott et al. (2015), Eq. (2) | Immersion | desert dust | $n_{250,\mathrm{d}}, T$ |
| U17-I(d) | Ullrich et al. (2017), Eq. (5) | Immersion | desert dust | $s_{\mathrm{d}}, T$ |
| U17-I(c) | Ullrich et al. (2017), Eq. (6) | Immersion | soot | $s_{\mathrm{c}}, T$ |
| U17-D(d) | Ullrich et al. (2017), Eq. (7) | Deposition | desert dust | $s_{\mathrm{d}}, T, S_{\mathrm{i}}$ |
| U17-D(c) | Ullrich et al. (2017), Eq. (7) | Deposition | soot | $s_{\mathrm{c}}, T, S_{\mathrm{i}}$ |

**Table 2.** Overview of ICNC and INPC retrieval results for the three closure studies on 10 April 2017 (CyCARE), and 8 and 17 March 2015 (Cyprus-2015 campaign). The vertical ice crystal number flux is the most direct observation and determined from the combined information of terminal velocity of the falling ice crystals, the radar reflectivity, and the lidar backscatter coefficient. ICNC is obtained from the ratio of the ice crystal number flux to the terminal velocity. The listed INPC values are maximum values found within the upper part of the cloud layer, i.e., at 8.4 km height on 10 April, 9.8 km on 17 March, and at 5.8 km height on 8 March 2015. All INPC and ICNC values are given with an uncertainty range of about a factor of 3 (in parentheses).

| Parameter | 10 April 2017 (Thin cirrus) | 8 March 2015 (Mixed-phase AC) | 17 March 2015 (Thick cirrus) |
|---|---|---|---|
| ICNC in $L^{-3}$ | 13 (4.3-39) | 1.4 (0.5-4.2), 2.5 (0.8-7.5) | 5 (1.7-15), 16 (5.3-48) |
| Ice crystal number flux in $m^{-2}s^{-1}$ | 3333 | 350, 750 | 1400, 4480 |
| Ice crystal terminal velocity in $m\ s^{-1}$ | 0.29 | 0.25, 0.3 | 0.28 |
| INPC in $L^{-3}$ , dust, U17-D, $S_i$=1.35 | 49.7 (16.6-149.1) | – | – |
| INPC, dust, U17-D, $S_i$=1.25 | 9.3 (3.1-27.9) | – | – |
| INPC, dust, U17-D, $S_i$=1.15 | 0.94 (0.31-2.82) | – | 67.5 (22.5-202.5) |
| INPC, dust, U17-D, $S_i$=1.10 | 0.23 (0.07-0.69) | – | 8.6 (2.9-25.8) |
| INPC, dust, U17-D, $S_i$=1.07 | – | – | 1.65 (0.55-4.95) |
| INPC, dust, U17-D, $S_i$=1.05 | 0.017 (0.006-0.051) | – | 0.4 (0.14-1.2) |
| INPC, dust, U17-I | – | 6.30 (2.1-18.9) | – |
| INPC, dust, D10(d) | – | 1.05 (0.35-3.15) | – |
| INPC, dust, D15 | – | 0.13 (0.04-0.39) | – |
| INPC, non dust, U17-I | – | 0.10 (0.03-0.3) | – |

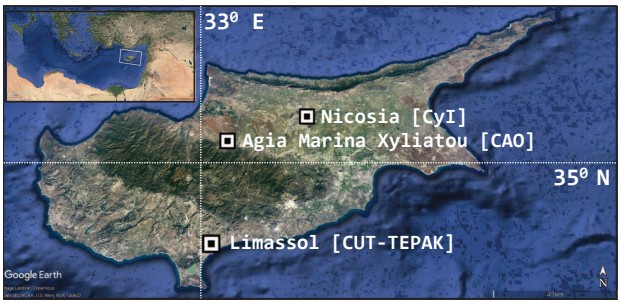

**Figure 1.** Map of Cyprus with the main field sites of the Cyprus-2015 and CyCARE campaigns. The inset shows the greater area of the Eastern Mediterranean, Middle East, and North Africa.

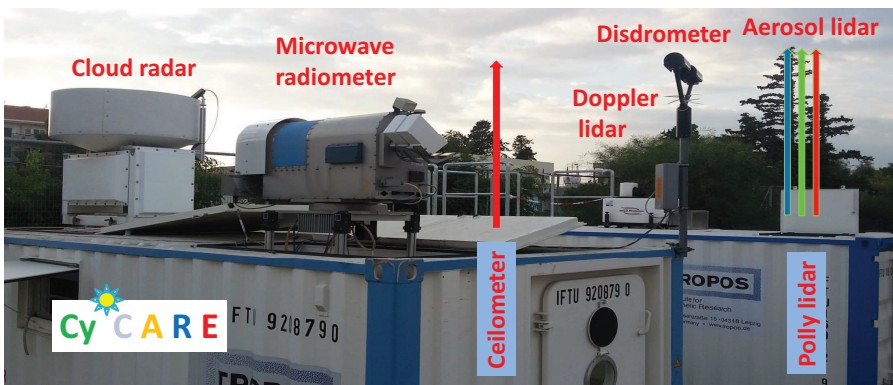

**Figure 2.** LACROS (Leipzig Aerosol and Cloud Remote Observations System) deployed at Limassol, Cyprus, from 21 October 2016 to 26 March 2018 in the framework of the CyCARE field campaign. LACROS belongs to Cloudnet and consists of a 35 GHz Doppler cloud radar, microwave radiometer, and disdrometer (on the roof of the cloud container), and a 1.5 $\mu$m wind Doppler lidar on the roof of the aerosol container and a multiwavelength polarization/Raman lidar (PollyXT) within the aerosol container. A ceilometer (1064 nm wavelength) was deployed between the cloud and aerosol containers (not visible in the photograph).

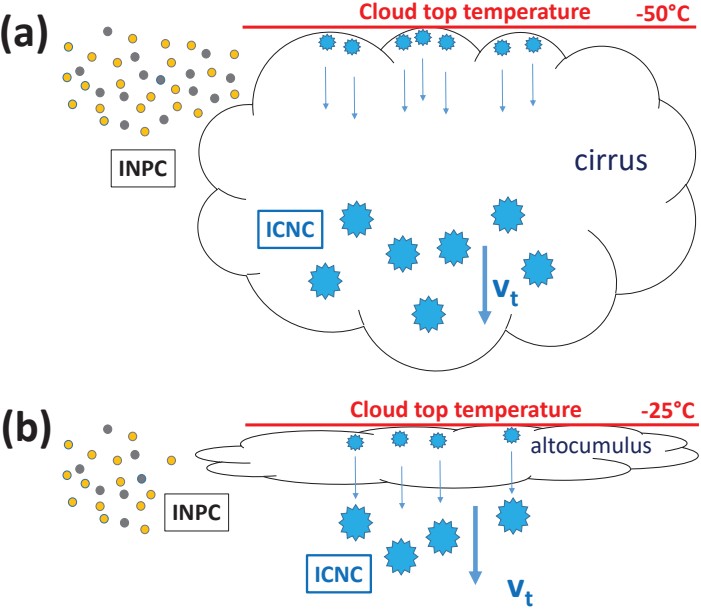

**Figure 3.** Illustration of the INPC-vs-ICNC closure approach for (a) a vertically deep cirrus cloud and (b) a shallow stratiform mixed-phase altocumulus layer. In the first step, INPC is retrieved from polarization lidar observations outside the cloud layer (e.g., before the cloud layer crosses the lidar field site). Gray and orange particles indicate continental pollution and mineral dust particles, respectively. In the second step, ICNC is estimated from observations with the available backscatter lidar, Doppler lidar, and cloud radar instrumentations. Within the cirrus layer (panel a) ICNC is preferable estimated in the lower part of the cloud with large ice crystals and thus clear ice crystal falling features. $v_t$ is the mean terminal velocity of the ice crystal population relative to the surrounding air. In the case of a shallow altocumulus layer (panel b), ICNC is determined in the ice virga zone, below the main cloud layer in which backscatter by liquid droplets dominate the lidar return signals and prohibit ice observations. In the closure experiments, we assume that all ice crystals nucleate at the top of the cloud layer (at the coldest point of the cloud layer), grow quickly and start falling, and that the number of nucleated ice crystals does not change on the way downward towards the lower part of the cirrus (panel a) and below the altocumulus layer (panel b), i.e., that collisions of ice crystals and subsequent formation of crystal aggregates as well as secondary ice production do not reduce the number of nucleated ice crystals significantly.

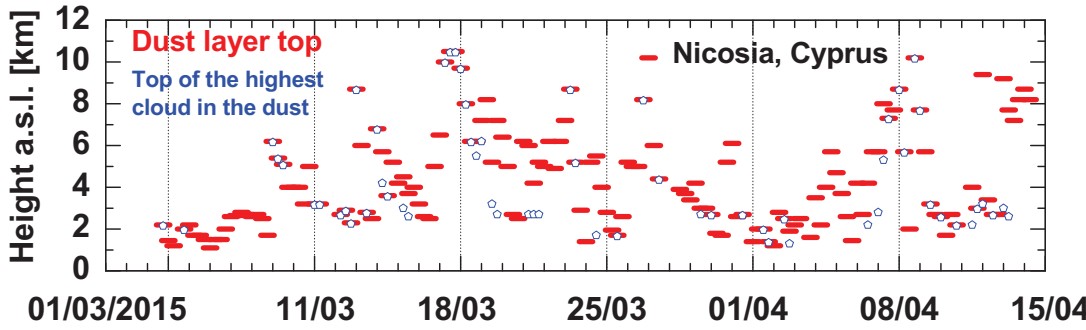

**Figure 4.** Dust and cloud conditions observed with lidar over Nicosia, Cyprus, during the BACCHUS Cyprus-2015 campaign from 1 March to 15 April 2015. The Polly lidar was running continuously over the six-week period. Six-hour periods (0–6, 6-12, 12-18, and 18-24 UTC) were evaluated separately regarding occurrence of dust, the top height of the uppermost dust layer (red short lines), cloud occurrence in dust and top height of the detected uppermost cloud layer (blue open symbols). Frequently clouds developed in Saharan dust, preferably in the top region of the dust layer.

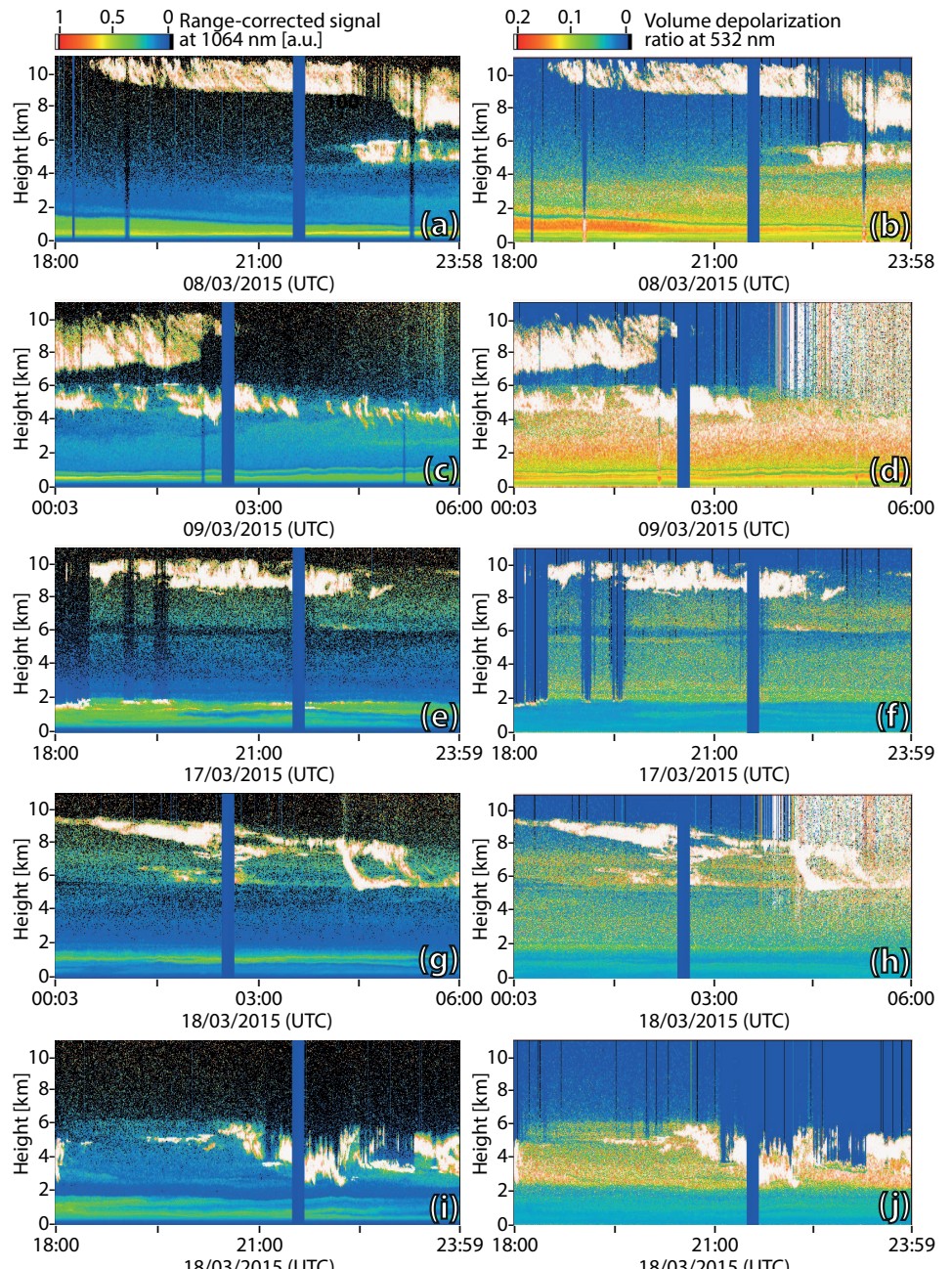

**Figure 5.** Cirrus and mixed-phase altocumulus layers (white areas) embedded in Saharan dust observed with polarization lidar during the BACCHUS Cyprus-2015 campaign. The range-corrected 1064 nm signal (left side, a,c,e,g,i) and the 532 nm volume depolarization ratio (right side, b,d,f,h,j) are shown. The depolarization ratio is especially sensitive to irregularly shaped particles such as mineral dust particles (values from 0.08 to 0.2, green to red) and ice crystals (>0.2, white). A remarkable coincidence of cloud layers with dust layers is visible which indicates a close link between dust occurence and ice formation. The thin vertical colums (dark because of low signal strength) are caused by strongly light-attenuating liquid-water clouds and near-surface fog. The well-defined broad vertical columns (around 2:30 and 21:40 UTC) indicate automatically performed polarization lidar calibration units.

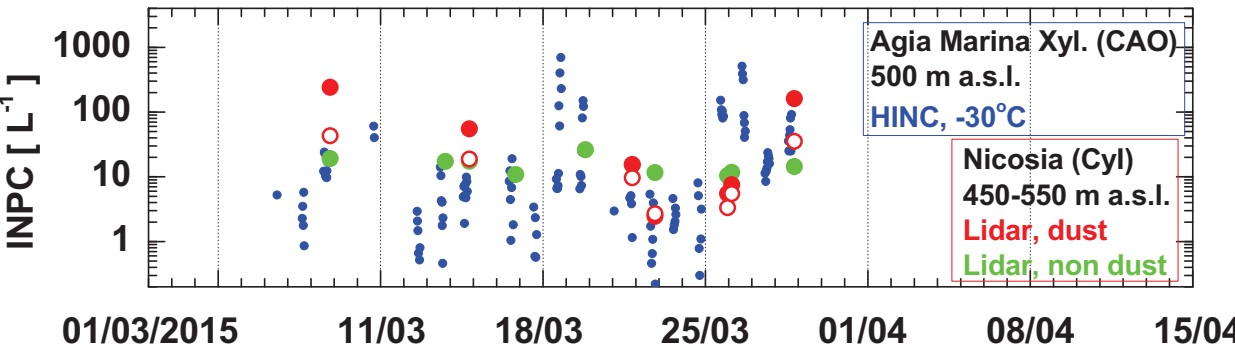

**Figure 6.** Comparison of in situ observations of INPC (HINC, ETH Zurich, see text for more explanations) at Agia Marina Xyliatou (about 500 m a.s.l.), and lidar-based INPC estimation (mean values for the 450-550 m a.s.l. height range) at Nicosia, about 25 km downwind of Agia Marina Xyliatou. All in situ observations collected in March 2015 are shown. The green circles indicate lidar estimates of INPC based on the D10 parameterization (for non-dust continental aerosol pollution) and the red circles are computed by using the D15 parameterization (for the dust aerosol fraction). For comparison, we also include dust INPC values (open red circles) obtained with the D10(d) parameterization (see Table 1). All INPC values (HINC, lidar) are given for $-30°$C.

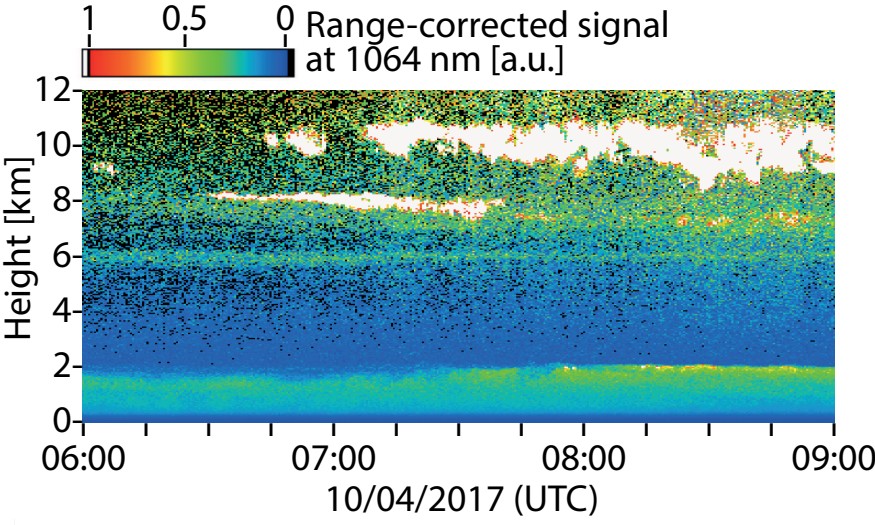

**Figure 7.** Development of a thin cirrus at 8 km height at 6:30 UTC on 10 April 2017. The thin cirrus layer was embedded in Saharan dust. An extended cirrus field is visible from 9–11 km height. The polluted boundary layer reached to about 2 km height. Ranged-corrected 1064 nm lidar signals are shown with 7.5 m and 30 s resolution.

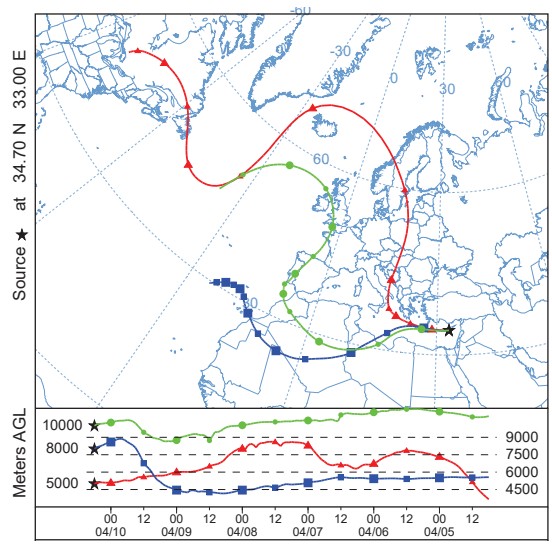

**Figure 8.** Six-day HYSPLIT backward trajectories arriving at Limassol, Cyprus, on 10 April 2017, 6:00 UTC. The backward trajectories are computed with the HYSPLIT (Hybrid Single Particle Lagrangian Integrated Trajectory) model (HYSPLIT, 2019; Stein et al., 2015; Rolph et al., 2017) and are based on GDAS0.5 meteorological fields. Arrival heights are 5000 m (red, clear air), 8000 m (blue, Saharan dust layer), and 10000 m (green, dust free upper troposphere). See Fig. 7 for comparison.

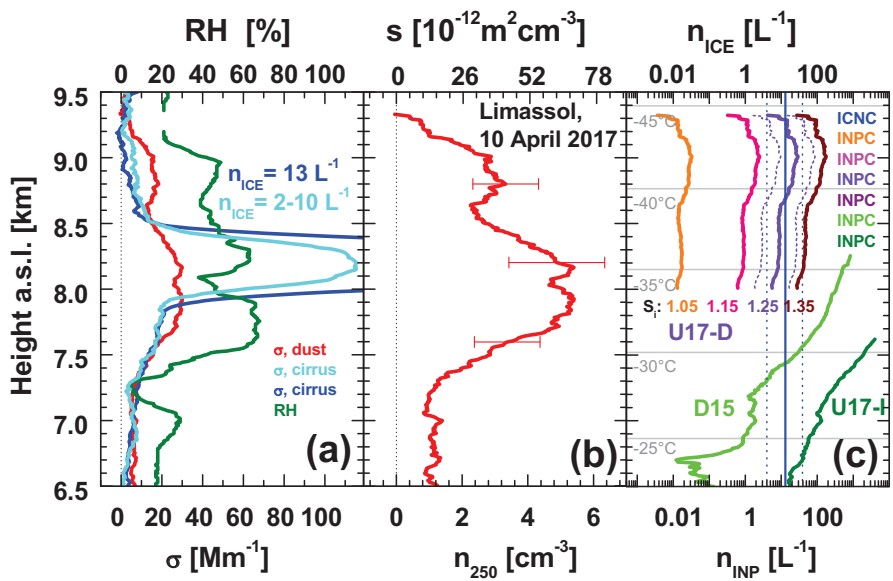

**Figure 9.** Cirrus INPC-vs-ICNC closure study based on LACROS observations at Limassol, Cyprus, on 10 April 2017 (see Fig. 7) during CyCARE. (a) Dust extinction coefficient (6:10-6:30 UTC mean), ice crystal extinction coefficient (light blue, 6:35-6:42 UTC mean, blue, 6:48-6:57 UTC mean, cirrus from 8–8.5km height), relative humidity (RH, radiosonde, launch at 5:50 UTC), estimated ICNC ($n_{ICE}$, light blue for the 6:35-6:42 UTC time period, blue for the 6:58-7:05 UTC time period), (b) dust particle number concentration $n_{250,d}$ (considering large particles with radius>250 nm only) and surface area concentration $s_d$ (6:10-6:30 UTC mean values), and (c) INPC ($n_{INP}$, 6:10-6:30 UTC mean) by using the deposition-nucleation U17-D(d) parameterization for four different ice supersaturation values ($S_i$ from 1.05-1.35). The uncertainty range of each of the four deposition-nucleation INPC profiles is indicated for $S_i$=1.10. For comparison, also immersion-freezing INPC profiles estimated by using the U17-I(d) (dark green) and D15 INP paramerizations (green, see Table 1) are shown. The ICNC value of 13 L$^{-1}$ (see the number in panel a) is shown as vertical blue line with the uncertainty range as vertical blue dashed lines.

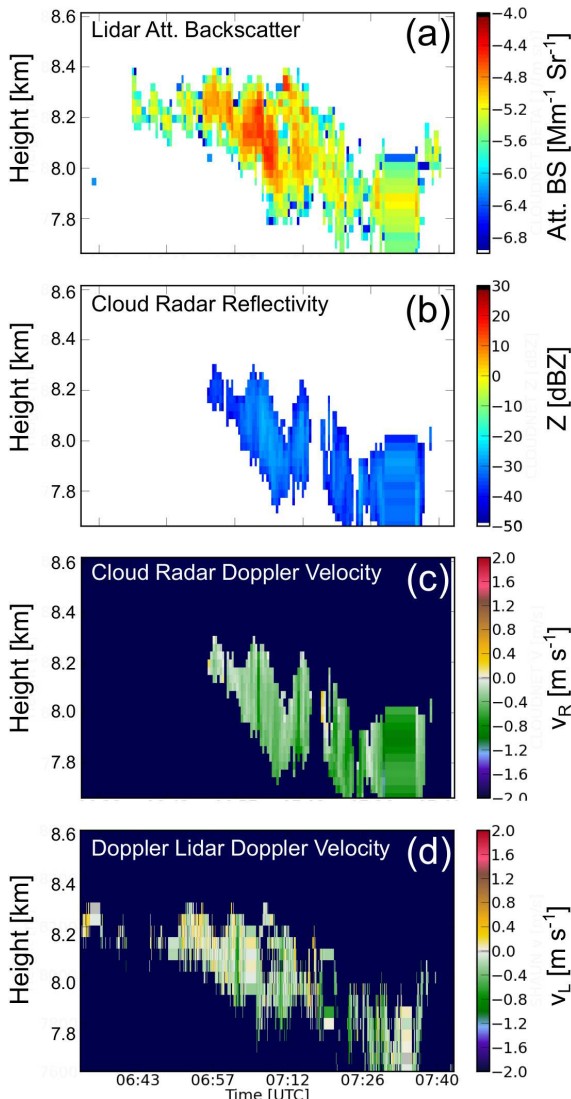

**Figure 10.** Evolution of a thin cirrus layer on 10 April 2017 (see also Fig. 7) as observed with Polly lidar in terms of (a) attenuated backscatter (Att. BS at 532 nm), with cloud radar in terms of (b) radar reflectivity $Z$ at 8.5 mm wavelength and (c) vertical velocity $v_R$, and with Doppler lidar in terms of (d) vertical velocity $v_L$.

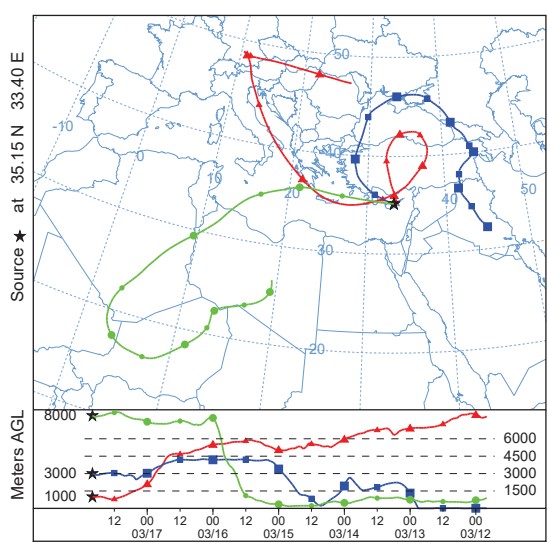

**Figure 11.** Six-day HYSPLIT backward trajectories arriving at Nicosia, Cyprus, on 17 March 2015, 20:00 UTC (see Figs. 5e and f). The computations are based on GDAS0.5 meteorological fields. Arrival heights are 1000 m (red, boundary layer), 3000 m (blue, Middle East dust layer), and 8000 m (green, Saharan dust layer).

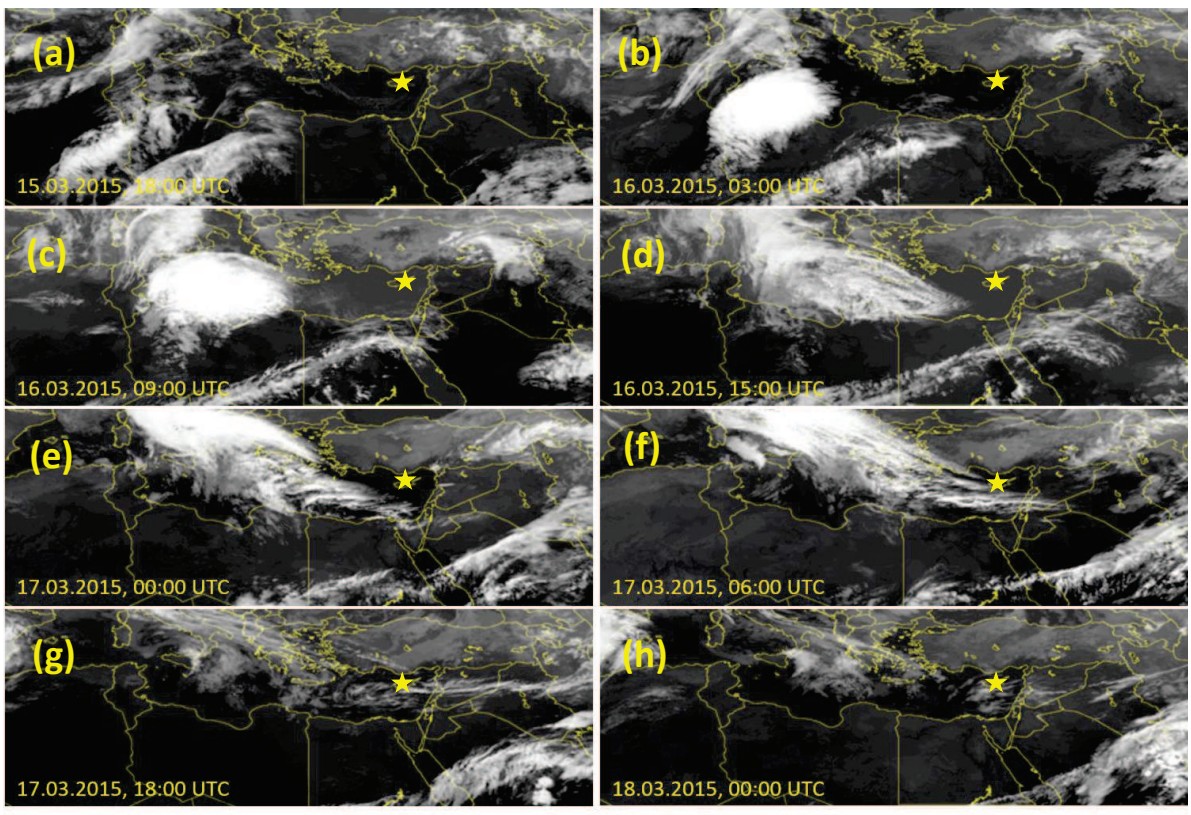

**Figure 12.** Evolution of a mesoscale convective system MCS (also denoted as dust-infused baroclinic storm, DIBS) (Fromm et al., 2016) over eastern Algeria and the central Mediterranean Sea on 15 March (a-c), followed by the decay and dissolution of the anvil cirrus shield on 16 March (d-f) and transformation into an active cirrus uncinus field extending from Crete to central Asia on 17 March 2019 (f-g). The lidar site at Nicosia is indicated by a yellow star. The Meteosat Second Generation (MSG) infrared 10.8 $\mu$m channel is used.

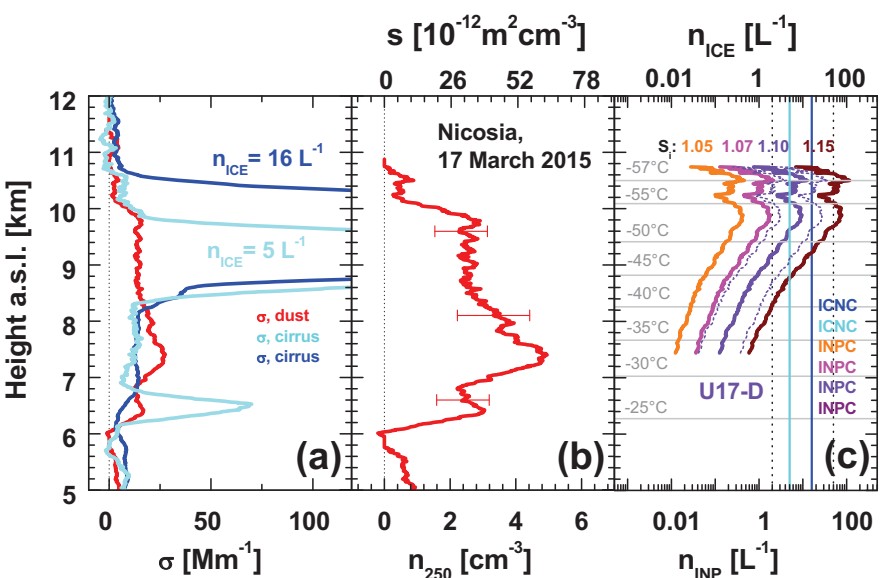

**Figure 13.** Same as Fig. 9 except for a cirrus layer observed with lidar over Nicosia on 17 March 2015 (see Figs. 5e and f) during the Cyprus-2015 campaign. (a) Dust extinction coefficient (23:05-23:30 UTC mean), ice crystal extinction coefficient (blue, 20:00-20:40 UTC mean, cirrus from 8.6-10.6 km, and light blue, 22:15-22:19 UTC mean, cirrus from 8.4-9.8 km), corresponding estimated ICNC ($n_{ICE}$ for the 20:00-20:40 and 22:15-22:19 UTC periods), (b) dust particle number concentration $n_{250,d}$ and surface area concentration $s_d$ (23:05-23:30 UTC mean values), and (c) INPC ($n_{INP}$, 23:05-23:30 UTC mean) by using the U17-D(d) parameterizations for four different ice supersaturation values ($S_i$ from 1.05-1.15). The uncertainty range of each of the four profiles is indicated for $S_i$=1.10 (violet thin dotted lines). The ICNC values (numbers in panel a) are shown as vertical light blue and dark blue lines with the overall uncertainty range as vertical black dashed lines.

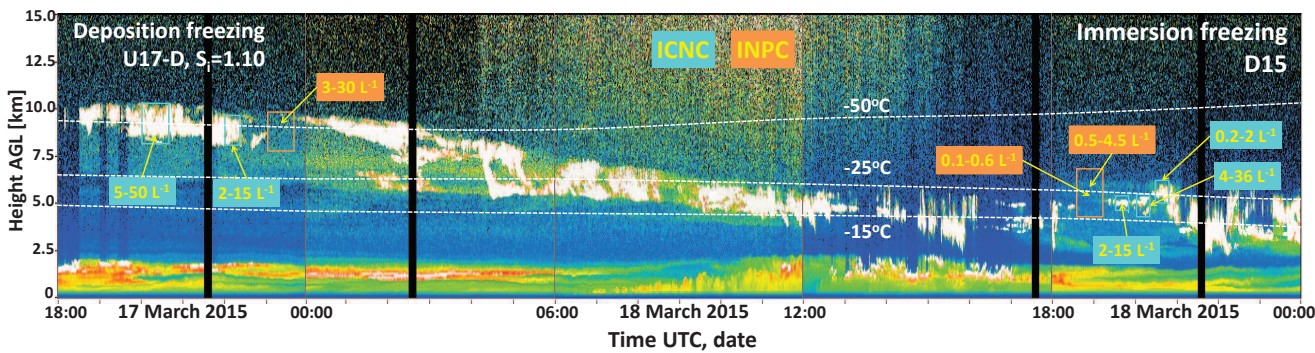

**Figure 14.** 30 hours of continuous cirrus and mixed-phase cloud observations over Nicosia on 17-18 March 2015 (also shown in Figs. 5e, g, and i). The air mass from 5-10 km height was replaced (starting at great heights) by dust-free, dry air advected from Turkey and southern Europe between 2:00 and 11:00 UTC on 18 March, leading to the impression of a descending dust and cirrus layer. Several INPC and ICNC values estimated from the lidar observations are given as numbers determined for the indicated orange (INPC) and blue (ICNC) boxes. The deposition-nucleation U17-I(d) parameterization is used on 17 March (at 9-10 km height for $S_i$=1.1) and the immersion-freezing D15 parameterization is applied in the evening data analysis on 18 March (at 5-6 km height). Dashed white lines show the GDAS1 temperature isolines with 3-hour resolution.

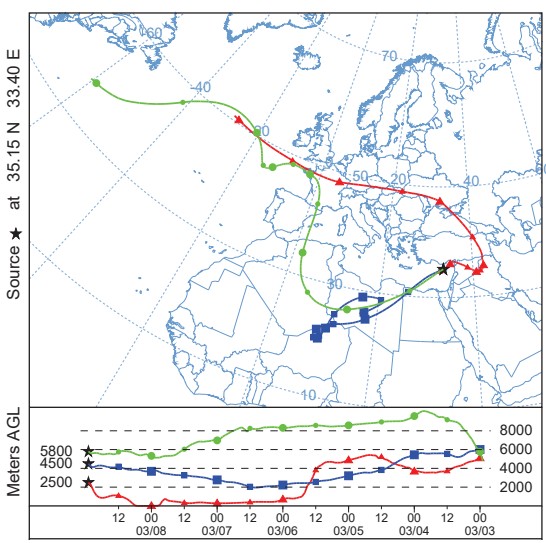

**Figure 15.** Six-day HYSPLIT backward trajectories arriving at Nicosia, Cyprus, on 8 March 2015, 23:00 UTC (see Figs. 5a and b). The computations are based on GDAS0.5 meteorological fields. Arrival heights are 2500 m (red, Middle East dust layer), 4500 m (blue, Saharan dust layer), and 5800 m (green, Saharan dust layer).

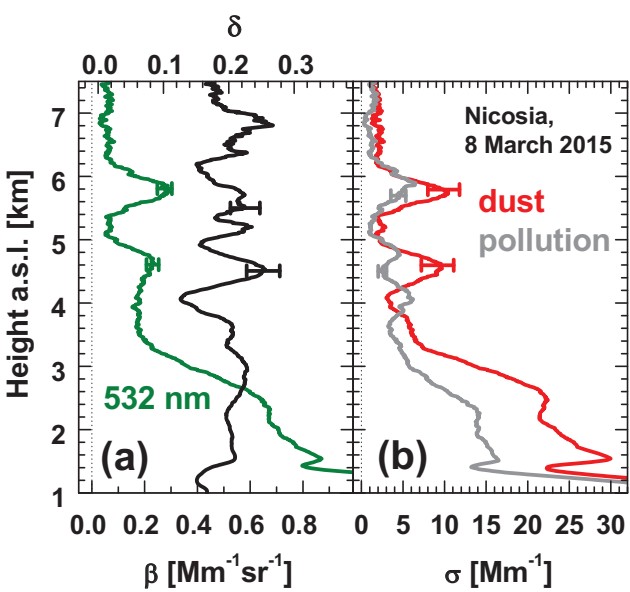

**Figure 16.** (a) Particle backscatter coefficient (green, 532 nm) and particle linear depolarization ratio (black, 532 nm), (b) corresponding dust and non-dust continental particle extinction coefficients. Mean lidar profiles for the time epriod from 22:00-22:20 UTC are shown. The uncertainty bars show typical retrieval uncertainties of 10% (backscatter, depolarization ratio) and 20% (extinction coefficient).

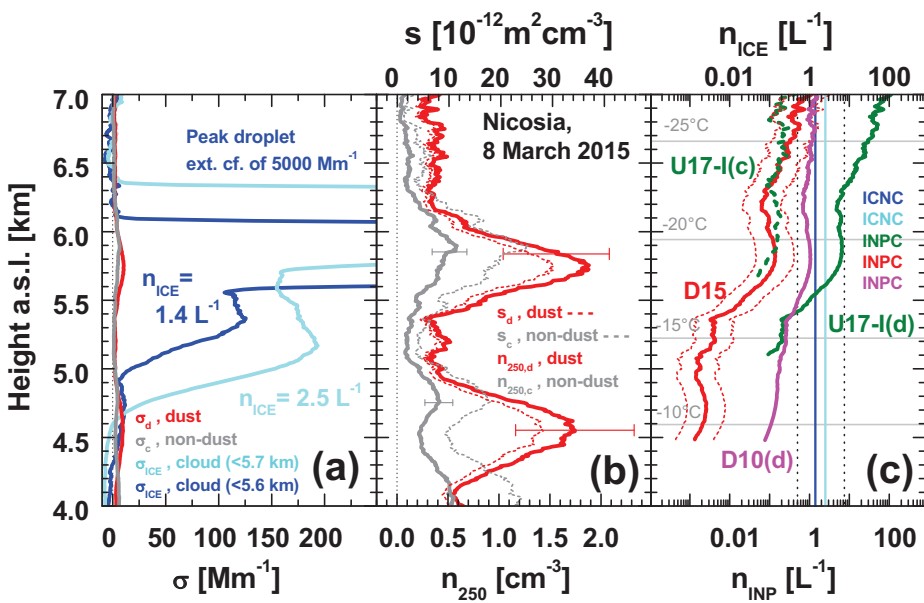

**Figure 17.** Mixed-phase-cloud INPC-vs-ICNC closure study based on lidar observations over Nicosia on 8 March 2015 (see Figs. 5a and b). (a) Dust (red) and non-dust (grey) particle extinction coefficient (22:00-22:20 UTC mean values, as shown in Fig. 16b), ice crystal extinction coefficient (blue, 22:29-22:31 UTC mean, ice virga from 5-5.6 km, and light blue, 22:43-22:53 UTC mean, ice virga from 4.7-5.7 km), estimated corresponding ICNC values ($n_{\mathrm{ICE}}$ in blue and light blue for the two time periods), (b) dust (red) and non-dust (grey) particle number concentration $n_{250,\mathrm{d}}$ and $n_{250,\mathrm{c}}$ (thick lines) and surface area concentration $s_{\mathrm{d}}$ and $s_{\mathrm{c}}$ (thin dashed lines, 22:00-22:20 UTC mean values), and (c) INPC ($n_{\mathrm{INP}}$, 22:00-22:20 UTC mean) by using the immersion-freezing INP parameterizations U17-I(c) (dotted dark green) for non-dust continental aerosol, and D10(d) (magenta, thick solid), D15 (red, thick solid, cf=1.0), and U17-I(d) (dark green, thick solid) for the dust aerosol fraction. The uncertainty range (factor 3) of each of the four INPC profiles is indicated for D15 only (red dotted lines). The ICNC values (numbers in panel a) are shown again, but here as vertical dark blue and light blue lines with the overall uncertainty range as vertical black dashed lines.