# Peer review of "Ice-nucleating particle versus ice crystal number concentration in altocumulus and cirrus embedded in Saharan dust: A closure study"

_Atmospheric Chemistry and Physics, 2019_

## Referee Comment (RC1) · Anonymous Referee #1 · 19 Jun 2019

General comments:
The novel algorithm to derive ice number concentrations from lidar observations on field sites and the comparison with empirical INP parameterization functions is shown in this manuscript. For three case studies covering both mixed-phase and ice clouds, the authors show the observed meteorological situation and vertical profiles. The comparison in the vertical profiles show a good agreement between both approaches. Additionally, the authors show the comparison of INP concentration from INP measurements on the surface level (HINC) and the lidar-derived concentrations. Since this is often a point of discussion, if surface-based techniques can represent INP concentration at cloud levels, the study confirms for this field campaign in Cyprus that

this is the case.

The reader is guided well through the topic itself und the different case studies. I have only some very minor comments for this nice and well written manuscript.

Specific comments:

- Sec. 5.3: I feel the need of a more detailed discussion of Fig.16 (c), e.g. U17-I(d) overestimates n_ice in the upper level, . . .

Technical corrections:

- p.2, l.4: ". . . ice formation (. . .) is not be possible . . ."

- p.3, l.23: Sect.3 is mentioned twice

- p.9, l.32: The superscripts on diameter initially looked to be like footnotes. Maybe it is better here to use a symbol instead.

- p.20, l.14: cloud

- Fig.16 (a): $\sigma$ subscripts should be mixed-phase here instead of cirrus

---

## Referee Comment (RC2) · Anonymous Referee #2 · 5 Jul 2019

General Comments:

It was a pleasure to review this "first time" closure study on the relationship between ice nucleating particle concentration (INPC) and ice crystal number concentration (ICNC) for two cirrus cloud closure studies and one altocumulus closure study, using state-of-the-science ground based remote sensing. These three successful closure studies validate part of the basic theory of ice cloud formation processes and thus make an important contribution to ice cloud and climate science research. The study was well conceived and executed, clearly relating the dust-enriched aerosol layer containing INPs with the cloud layer in time and space (conditions that strongly favor heterogeneous

ice nucleation, or het). The contribution of mineral dust and air pollution (non-dust) aerosol were addressed for deposition and immersion freezing ice nucleation. The paper is well written and organized, showing attention to important details. Sufficient detail and references are given for other investigators to attempt similar studies. This study provides a basis for more advanced field studies on ice nucleation.

The main drawback to this study (as far as this reviewer can see) is the assumption that ICNC can be estimated in the lower part of a cirrus cloud (see Fig. 3). Retrieved vertical ICNC distributions from cirrus (Mitchell et al., 2018, APC, Fig. 10) show N $\sim$ 5 times higher near cloud top relative to the lower half of a cirrus cloud for conditions likely associated with het (e.g. over mid-latitude oceans during summer). Aircraft measurements show ice nucleation is most frequent near cloud top (Diao et al., 2015, JGR). Kanji et al. (2017) show that INPC strongly increases with decreasing temperature for most environmental conditions. Together, these studies indicate ICNC from het is generally much higher near cloud top where ice nucleation prevails, and that relating INPC to ICNC near cloud base is misleading, underestimating ICNC by perhaps a factor of 5.

The modeling work of Spichtinger and Gierens (2009a & b, ACP) for a cirrostratus case study also indicates that ICNC is not quasi-constant throughout the cirrus deck, but is highest near cloud top. This is due to the relative humidity wrt ice (RHi) being highest near cloud top, since freshly nucleated ice crystals grow fast and fall, removing ice surface area from this region, thus allowing RHi (supplied by the constant updraft) to remain relatively high near cloud top. The falling ice crystals deplete RHi lower in the cloud, reducing the rate of ice nucleation there (mostly due to homogeneous ice nucleation in this study, but it may apply to het as well since het also depends on RHi). Although there is a constant supply of new ice crystals near cloud top, these are spread vertically over the whole cloud depth, thus diluting the ICNC near cloud top in the lower cloud.

These concerns are only relevant to the second case study (17-18 March 2015) where

the cirrus cloud layer can be up to 2 km deep or more. In the first case study (10 April 2017), the cirrus layer was probably too thin for this phenomena to manifest, and for the altocumulus case study, this cloud was quite thin and mixed phase, where the above arguments and observations do not apply.

To summarize, these factors should be considered when evaluating the results from the second INPC-ICNC closure experiment. Even better would be to retrieve ICNC near cloud top during this experiment. Since successful closure is defined as agreement within an order of magnitude regarding INPC and ICNC, successful closure may still be achieved, but the authors may wish to sample ICNC and frame their arguments somewhat differently to accommodate these points.

Finally, the central findings of this study would be more accessible to a casual reader if a table were included (similar to Table 2) that shows the main results (i.e. INPC vs. ICNC) for each of the three closure experiments (i.e. all main closure experiment results in a single table), provided this does not oversimplify the findings too much. In this way, one will not need to search through the text to find these key results.

Major Comments:

1. Page 3, lines 2-4: Please relate this to the above discussion.

2. Page 3, line 14: For completeness, please also cite the satellite retrieval technique of Mitchell et al. (2018, ACP) that retrieves ICNC, De and IWC.

3. Page 3, lines 17-18: This is true for polar orbiting satellites but not for geostationary satellites; please qualify this sentence.

4. Page 7, lines 29-31, and page 8, lines 1-5: Please revise this in accordance with the "General Comments" section.

Minor Comments:

1. Page 1, line 4: "extend" => extent?

**[ACPD](...)**

Interactive
comment

2. Page 2, lines 9-10: Because aerosols play a role in the tropospheric water cycle, does this imply that this water cycle is sensitive to aerosols (e.g. their chemistry and concentration)? For example, over the typical range of CCN and INP concentration, is the water cycle sensitive to changes in their concentration? Or are there generally sufficient CCN and INP to accommodate the vertical transport of water?

3. Page 12, line 9: Unbalanced parentheses

4. Page 19, line 30: "extend" => extent?

5. Page 20, line 14: "cloid" => cloud?

---

## Referee Comment (RC3) · Anonymous Referee #2 · 10 Jul 2019

Addendum:

I forgot to include a comment about the calculation of ICNC based on the quotient of the vertical ice crystal number flux (most direct observation) and the ice crystal terminal velocity vt (as stated in Table 2). More information about vt is needed. For example, vt should be based on the ice particle size distribution (PSD), and can be either number- or mass-weighted. Since ICNC is based on the number flux, vt should be the number-weighted ice fall speed:

$$vN = \sum v(D)N(D)\Delta D / \sum N(D)\Delta D = \sum v(D)N(D)\Delta D / N,$$

where D = ice crystal maximum dimension, v(D) is the fall speed, N(D) is the PSD, N is

the total PSD number concentration and $\Delta D$ is the size bin width of the PSD. Doppler lidar estimates of vN should be adequate, but estimating vN from radar reflectivity and the ice cloud extinction coefficient appears more dubious (Sect. 4.2.2). Please address this concern.

---

## Short Comment (SC1) · 10 Jul 2019

One of the case studies in this manuscript compelled me to offer some informal comments. A small group of scientists have been studying what we're calling dust-infused baroclinic storms (DIBS). What we found has been published in two papers.

https://agupubs.onlinelibrary.wiley.com/doi/full/10.1002/2016GL071801

https://agupubs.onlinelibrary.wiley.com/doi/full/10.1029/2017JD027848

The DIBS phenomenon, I believe, bears directly on one of Ansmann et al.'s case study of 17 March 2015. In particular, the back trajectory from the dusty cirrus layer reveals

some critical aspects of the air mass history that may alter the paper's interpretation of the cloud and aerosol observed over Cyprus. The back trajectory indicates dramatic warm conveyor belt uplift prior to some hours of relatively level transport. We find in our cases that the RH along such trajectories also suggests cloud formation (RH increasing to >80%). When combined with geostationary and polar orbiting visible and IR imagery we were able to show that the ascending dust essentially entered the cloud base of the baroclinic storm, flowed through the storm cloud from its lowest (liquid) levels up to ice-only temperatures, infusing the cloud wth nucleation sites. The storm itself exhibits lots of peculiar physical and microphysical signals consistent with dust infusion all the way to the cloud top. The 17 March cirrus observation case is preceded by the "classic"DIBS signatures laid out in the above-referenced papers. E.g. the DIBS cirrus deck has a peculiar cellular texture (for reasons we still are trying to understand). Here's a view of the MODIS brightness temperature showing the cellular DIBS cloud north of Libya on 16 March, right along the Figure 11 back trajectory.

https://go.nasa.gov/2EjHvBL

By 17 March the residual DIBS cloud had expanded over southern Europe, all the way to Cyprus.

https://go.nasa.gov/2Eld9yy

My guess is that SEVIRI visible and IR imagery would enable a full reconstruction of that DIBS from formation on ~15 March to the time of the 17 March observations. And unless I am letting my eyes fool me, I think I can discern a cellular nature in the lidar time series in Figure 5e-f (especially the depolarization ratio). We've seen the cellular structure in other lidar/radar depictions of DIBS. Here is an example:

https://www.eumetsat.int/website/home/Images/ImageLibrary/DAT_IL_10_02_11.html

The characterization in Ansmann et al. of this as a "long-lasting" cirrus cloud is consistent with the synoptic view of this DIBS cloud. According to the back trajectory (Fig.

11), it might be argued that one is seeing an air mass involving ice and dust that traces back more than a day, when the air was flowing through the WCB and inside clouds of liquid and then ice.

One of the aspects of DIBS that the EUMETSAT folks have been documenting is a cloud lifetime effect of these "dusty cirrus."

https://www.eumetsat.int/website/home/Images/ImageLibrary/DAT_3008816.html

Maybe the long-lasting dusty cirrus over Cyprus on 17 March 2015 was also holding on longer than expected? I'd be curious to get the authors' thoughts on the DIBS perspective and whether it further informs their understanding of the ice-

–––––––––––––––––––––––––––

---

## Author Comment (AC1) · 7 Sep 2019

The comment was uploaded in the form of a supplement:
https://www.atmos-chem-phys-discuss.net/acp-2019-447/acp-2019-447-AC1-supplement.pdf

---

## Author Comment (AC2) · 7 Sep 2019

Dear Editor, dear reviewers!

The following letter includes our reply to all comments of the three reviewers.

We thank the reviewers for careful reading and for making good suggestions. We considered almost all of them. Our answers are in blue.

Before we provide an item by item reply, we provide an overview of main changes and improvements:

- The two tables are improved, and Table 2 now contains summarizing ICNC and INPC numbers for all three cases as requested
- The critical assumptions in our closure approach (Sect. 4) are better highlighted, e.g., that the impact of crystal collision and aggregation effects may lead to a factor of 3-10 lower ICNC values in the lower part of the cirrus in comparison to the nucleation-related ICNC values at cloud top. Respective consequences for the 17 March 2015 case are discussed in Sect. 5.2.
- The ICNC retrieval is now better explained. More details to the use of the terminal velocity information (number weighted approach) are given (Sect 4.2.2).
- Extended discussion on the occurrence of a very large mesoscale convective system (MCS), … 'yes it was a strong DIBS (as suggested by Mike Fromm)!' … is now included (Sect. 5.2). A new figure (Fig.12, MSG images, 10.8µm channel) is added to highlight this unusual case of complex cloud evolution. The new Fig. 12 shows the full life cycle of the cloud system from the formation of the MCS, the associated anvil cirrus, and the transformation into an active cirrus uncinus field (Sect. 5.2).
- The strong differences in the INPC estimation (for immersion freezing, D15 vs U17-I(d), almost a factor 50 difference, 8 March case study) are now discussed in large detail. Figure 17 is now more easy to understand because the D10 INPC profile for non-dust aerosol and the respective uncertainty profiles are removed.
- The summary and conclusion section is re-written and thus has now a better structure.

In the revised version of the manuscript which is also included in this document behind the reply letter, all significant changes are highlighted in **BOLD.**

**Reviewer #1**

General comments:

I have only some very minor comments for this nice and well written manuscript.

Specific comments:

• Sec. 5.3: I feel the need of a more detailed discussion of Fig.16 (c), e.g. U17-I(d) overestimates n_ice in the upper level.

This is now done (Sect. 5.3, Fig 17c now, page 20)!

Technical corrections:

• p.2, l.4: "… ice formation (…) is not be possible …"

Improved!

• p.3, l.23: Sect.3 is mentioned twice

Improved!

• p.9, l.32: The superscripts on diameter initially looked to be like footnotes. Maybe it is better here to use a symbol instead.

We agree! Improved!

• p.20, l.14: cloud

Improved!

• Fig.16 (a): _ subscripts should be mixed-phase here instead of cirrus

We use index 'ICE' now, and also give the height range for which the extinction coefficient shows ice extinction values in Fig. 17a (former 16a).

**Reviewer #2**

General Comments:

….The paper is well written and organized, showing attention to important details. Sufficient detail and references are given for other investigators to attempt similar studies. This study provides a basis for more advanced field studies on ice nucleation.

The main drawback to this study (as far as this reviewer can see) is the assumption that ICNC can be estimated in the lower part of a cirrus cloud (see Fig. 3). Retrieved vertical ICNC distributions from cirrus (Mitchell et al., 2018, APC, Fig. 10) show N approx. 5 times higher near cloud top relative to the lower half of a cirrus cloud for conditions likely associated with het (e.g. over mid-latitude oceans during summer). Aircraft measurements show ice nucleation is most frequent near cloud top (Diao et al., 2015, JGR). Kanji et al. (2017) show that INPC strongly increases with decreasing temperature for most environmental conditions. Together, these studies indicate ICNC from het is generally much higher near cloud top where ice nucleation prevails, and that relating INPC to ICNC near cloud base is misleading, underestimating ICNC by perhaps a factor of 5.

This is now discussed in large detail. But the main reason for the ICNC reduction with increasing distance from cloud top is probably ice-ice collision and aggregation (Field and Heymsfield, 2003, Field et al., 2006). This is now mentioned, and confirmed by CALIPSO lidar observations (Mitchell et al., 2008)… Sect. 4, page 8-9.

The modeling work of Spichtinger and Gierens (2009a & b, ACP) for a cirrostratus case study also indicates that ICNC is not quasi-constant throughout the cirrus deck, but is highest near cloud top. This is due to the relative humidity wrt ice (RHi) being highest near cloud top, since freshly nucleated ice crystals grow fast and fall, removing ice surface area from this region, thus allowing RHi (supplied by the constant updraft) to remain relatively high near cloud top. The falling ice crystals deplete RHi lower in the cloud, reducing the rate of ice nucleation there (mostly due to homogeneous ice nucleation in this study, but it may apply to het as well since het also depends on RHi). Although there is a constant supply of new ice crystals near cloud top, these are spread vertically over the whole cloud depth, thus diluting the ICNC near cloud top in the lower cloud.

Spichtinger and Gierens, 2009a (i.e., their part 1b) and 2009b (part 2) are now mentioned, but not in the details describe above. Only short! Section 4, page 8-9.

These concerns are only relevant to the second case study (17-18 March 2015) where the cirrus cloud layer can be up to 2 km deep or more. In the first case study (10 April 2017), the cirrus layer was probably too thin for this phenomena to manifest, and for the altocumulus case study, this cloud was quite thin and mixed phase, where the above arguments and observations do not apply.

Yes! … and this is also stated (page 8)!

To summarize, these factors should be considered when evaluating the results from the second INPC-ICNC closure experiment. Even better would be to retrieve ICNC near cloud top during this experiment. Since successful closure is defined as agreement within an order of magnitude regarding INPC and ICNC, successful closure may still be achieved, but the authors may wish to sample ICNC and frame their arguments somewhat differently to accommodate these points.

Yes we considered these effects in the discussion of findings! See Section 5.2, page 8!

Regarding our ICNC retrieval …. We cannot go into upper part of the cloud! …. It is problematic because (a) the crystals are smaller close to cloud top, and the terminal fall speed is lower, and (b) air motion (updraft, downdraft motion) is less easy to separate from ice crystal fall speed in the cloud top region… A separation would be possible in the case of additional wind profiler measurements of the air motion component, but we did not have a wind profiler at Cyprus. This is a future aspect, and given in the conclusions.

Finally, the central findings of this study would be more accessible to a casual reader if a table were included (similar to Table 2) that shows the main results (i.e. INPC vs. ICNC) for each of the three closure experiments (i.e. all main closure experiment results in a single table), provided this does not oversimplify the findings too much. In this way, one will not need to search through the text to find these key results.

We followed the reviewer's suggestion and improved and extended Table 2 accordingly.

Major Comments:

1. Page 3, lines 2-4: Please relate this to the above discussion.

Done! …. by providing the Spichtinger and Gierens (2009a, 2009b) and Field and Heymsfield (2003) references in Sect. 1 (page 3). However, we do not want to confuse the reader with too much side information already in the discussion, where we need a straight forward argumentation.

2. Page 3, line 14: For completeness, please also cite the satellite retrieval technique of Mitchell et al. (2018, ACP) that retrieves ICNC, De and IWC.

Done, and cited! Sect. 1, page 3.!

3. Page 3, lines 17-18: This is true for polar orbiting satellites but not for geostationary satellites; please qualify this sentence.

Improved!

4. Page 7, lines 29-31, and page 8, lines 1-5: Please revise this in accordance with the

"General Comments" section.

Improved! …as mentioned above…Sect.4, page 8-9.

Minor Comments:

1. Page 1, line 4: "extend" => extent?

Improved!

2. Page 2, lines 9-10: Because aerosols play a role in the tropospheric water cycle, does this imply that this water cycle is sensitive to aerosols (e.g. their chemistry and concentration)? For example, over the typical range of CCN and INP concentration, is the water cycle sensitive to changes in their concentration? Or are there generally sufficient CCN and INP to accommodate the vertical transport of water?

Improved! We changed the text a bit to make it more clear (Sect. 1, page 2). Only the physical role, …. acting as CCN and INP …., is meant. On the other hand, there is no room for long explanations here in this introduction on ICNC-vs-INPC closure.

3. Page 12, line 9: Unbalanced parentheses

Improved!

4. Page 19, line 30: "extend" => extent?

Improved!

5. Page 20, line 14: "cloid" => cloud?

Improved!

Addendum:

I forgot to include a comment about the calculation of ICNC based on the quotient of the vertical ice crystal number flux (most direct observation) and the ice crystal terminal velocity vt (as stated in Table 2). More information about vt is needed. For example, vt should be based on the ice particle size distribution (PSD), and can be either number or mass-weighted. Since ICNC is based on the number flux, vt should be the number weighted ice fall speed:

$$vN = P\ v(D)\ N(D)\ Delta\_D\ /\ P\ N(D)\ Delta\_D = P\ v(D)\ N(D)\ Delta\_D\ /\ N,$$

where D = ice crystal maximum dimension, v(D) is the fall speed, N(D) is the PSD, N is the total PSD number concentration and Delta_D is the size bin width of the PSD. Doppler lidar estimates of vN should be adequate, but estimating vN from radar reflectivity and the ice cloud extinction coefficient appears more dubious (Sect. 4.2.2). Please address this concern.

Improved! (see Section 4.2.2, page 11, and Fig. 3 caption).

We realized that the terminal velocity $v\_t$ was not properly defined in the text. Hence, this definition was added to the caption of Fig. 3 and in the text now. The influence of the different instrumental weightings on the measurement of $v\_t$ is taken into account during the ICNC retrieval (Bühl et al., 2019). This is now mentioned.

In principle, a broad variety of PSDs are simulated and observable variables (radar reflectivity, lidar backscatter, Doppler velocity for radar/lidar, etc…) are estimated from those distributions. By comparison of the simulated and the measured values, the correct PSD is then chosen. The shift between the "real" mean fall velocity of the particles and the fall velocities measured with lidar and radar is hence taken into account implicitly.

**Reviewer #3 (M. Fromm)**

One of the case studies in this manuscript compelled me to offer some informal comments. A small group of scientists have been studying what we're calling dust-infused baroclinic storms (DIBS). What we found has been published in two papers.

We discuss DIBS now in Section 5.2 and add these two papers to the references. The full cloud life cycle is now discussed as stimulated by Mike Fromm! He is completely right, and we changed to text accordingly by adding an extended description of the cloud evolution over Africa and dust uplift. We introduce a new figure with Meteosat images (Fig.12).

The DIBS phenomenon, I believe, bears directly on one of Ansmann et al.'s case study of 17 March 2015. In particular, the back trajectory from the dusty cirrus layer reveals some critical aspects of the air mass history that may alter the paper's interpretation of the cloud and aerosol observed over Cyprus. The back trajectory indicates dramatic warm conveyor belt uplift prior to some hours of relatively level transport.

Yes, we agree and discuss the full MCS/DIBS/anvil-cirrus/cirrus-uncinus complex in large detail supported by Fig. 12.

We find in our cases that the RH along such trajectories also suggests cloud formation (RH increasing to >80%). When combined with geostationary and polar orbiting visible and IR imagery we were able to show that the ascending dust essentially entered the cloud base of the baroclinic storm, flowed through the storm cloud from its lowest (liquid) levels up to ice-only temperatures, infusing the cloud with nucleation sites. The storm itself exhibits lots of peculiar physical and microphysical signals consistent with dust infusion all the way to the cloud top. The 17 March cirrus observation case is preceded by the "classic" DIBS signatures laid out in the above-referenced papers. E.g. the DIBS cirrus deck has a peculiar cellular texture (for reasons we still are trying to understand).

The cirrus observed over Cyprus however was a classical cirrus uncinus as defined by Sassen et al. (2002 in this Oxford 'Cirrus' book) and thus clearly different from the DIBS-related cirrus. The main reason is that the DIBS occurred much much earlier (early morning of 16 May), and no cirrus can maintain these unique features over almost 1.5 days (until the evening of 17 March). The anvil cirrus shield dissolved and transformed into an ordinary synoptic cirrus layer. This is described in the main text body. So, we did not observe a strong backscatter peak at cloud top (as described by Fromm et al., 2016, Caffrey et al., 2018), we did not have small crystals, and thus not large ice crystal number concentrations. However, because of the DIBS, there was humid air in the upper troposphere AND, and at the same time…., a lot of dust and thus optimum conditions for heterogeneous ice nucleation via deposition nucleation. And the unusually long cirrus life time and the rather large regional coverage are clear indications for the 'unlimited' impact of the unlimited source of INPs. All this is explained in Sect. 5.2 and also later in the conclusions.

By 17 March the residual DIBS cloud had expanded over southern Europe, all the way to Cyprus.

My guess is that SEVIRI visible and IR imagery would enable a full reconstruction of that DIBS from formation on _15 March to the time of the 17 March observations. And unless I am letting my eyes fool me, I think I can discern a cellular nature in the lidar time series in Figure 5e-f (especially the depolarization ratio). We've seen the cellular structure in other lidar/radar depictions of DIBS.

Cellular structures are regular features of typical cirrus layers, i.e., nucleation cells plus virga zones. Radiative cooling at cloud top can cause the almost regular occurrence of updrafts and downdrafts. And in the updraft zones the typical uncinus cells (ice nucleation cells) occur and create the typical uncinus signature and cirrus uncinus features.

The characterization in Ansmann et al. of this as a "long-lasting" cirrus cloud is consistent with the synoptic view of this DIBS cloud. According to the back trajectory (Fig. 11), it might be argued that one is seeing an air mass involving ice and dust that traces back more than a day, when the air was flowing through the WCB and inside clouds of liquid and then ice.

Maybe the long-lasting dusty cirrus over Cyprus on 17 March 2015 was also holding on longer than expected? I'd be curious to get the authors' thoughts on the DIBS perspective and whether it further informs their understanding of the ice.

See our reply above…

Thank you for exciting impact on the paper structure!

[revised manuscript text omitted]

---

## Author Comment (AC3) · 7 Sep 2019

The comment was uploaded in the form of a supplement:
https://www.atmos-chem-phys-discuss.net/acp-2019-447/acp-2019-447-AC3-supplement.pdf

---

## Author Comment (AC4) · 7 Sep 2019

The comment was uploaded in the form of a supplement:
https://www.atmos-chem-phys-discuss.net/acp-2019-447/acp-2019-447-AC4-supplement.pdf

---

## Author Response (AR2)

Dear ACP Editorial and Production Team

Below is a short list of the final improvements we included, before uploading the final files (text and figure files).

**The paper should now be (again) a BACCHUS Special Issue Contribution**

We submitted the paper to ACP as BACCHUS special issue contribution (end of March 2019). But no BACCHUS editor could be found within six weeks (until 5 May 2019). So, we saw no alternative…, and withdrew the manuscript. We re-submitted the paper at the same day (in May 2019) as a regular paper so that non-BACCHUS editors could take it. And that worked. Now the paper is accepted and recommended to be even a highlight paper.

Back to BACCHUS special issue: Because the article is one of the main papers with key BACCHUS results, it should be a BACCHUS Special Issue paper.

We inserted already the special issue statement… in the uploaded manuscript.

Final improvements (as suggested by the editor)

Mike Fromm's message to the authors:

Ansmann et al.'s responses to my review were totally satisfactory, with one exception. They refer to the Dust Infused Baroclinic Storm (DIBS) as a mesoscale convective complex (MCS). There is no basis that given in the DIBS papers they cite. The DIBS is a synoptic-scale dynamics driven storm, not a complex of thunderstorms. I would hope that Ansmann et al. will either remove the MCS references, or make their own argument for it.

***We agree and changed accordingly!***

Reviewer #2

The authors have satisfactorily addressed all my concerns and the paper is now almost ready for publication in ACP in my view. Some minor improvements are suggested below.

Page 4, lines 8-10: This could reduce cirrus cloud lifetimes by changing the ice nucleation mechanism from homo- to heterogeneous; see, for example, Storelvmo et al. (2014, Phil. Trans. Royal Met. Soc.) and Gruber et al. (2019, JGR).

Page 22, line 22: prpfiling => profiling?

Page 23, line 9: Two typos here.

***We removed the typos and mentioned the statement (reduced life cycle and give the suggested references) in the introduction (page 4).***